# Structural and Functional Biology of Mammalian ALOX Isoforms with Particular Emphasis on Enzyme Dimerization and Their Allosteric Properties

**DOI:** 10.3390/ijms252212058

**Published:** 2024-11-09

**Authors:** Alexander Zhuravlev, Viktor Gavrilyuk, Xin Chen, Vladislav Aksenov, Hartmut Kuhn, Igor Ivanov

**Affiliations:** 1Lomonosov Institute of Fine Chemical Technologies, MIREA—Russian Technological University, Vernadskogo pr. 86, Moscow 119571, Russia; alekszhur95@yandex.ru (A.Z.); aksenov.v.v@edu.mirea.ru (V.A.); ivanov_i@mirea.ru (I.I.); 2Department of Biochemistry, Charite, University Medicine Berlin, Corporate Member of Free University Berlin, Humboldt University Berlin and Berlin Institute of Health, Charitéplatz 1, D-10117 Berlin, Germany; xin.chen2@charite.de

**Keywords:** eicosanoids, lipoxygenase, allosteric enzymes, structural flexibility, protein dimerization

## Abstract

The human genome involves six functional arachidonic acid (AA) lipoxygenase (*ALOX*) genes, and the corresponding enzymes (ALOX15, ALOX15B, ALOX12, ALOX12B, ALOXE3, ALOX5) have been implicated in cell differentiations and in the pathogenesis of inflammatory, hyperproliferative, metabolic, and neurological disorders. Humans express two different AA 15-lipoxygenating ALOX isoforms, and these enzymes are called ALOX15 (15-LOX1) and ALOX15B (15-LOX2). Chromosomal localization, sequence alignments, and comparison of the enzyme properties suggest that pig and mouse ALOX15 orthologs (leukocyte-type 12-LOX) on the one hand and rabbit and human ALOX15 orthologs on the other (reticulocyte-type 15-LOX1) belong to the same enzyme family despite their different reaction specificities with AA as a substrate. In contrast, human ALOX12 (platelet-type 12-LOX), as well as pig and mouse ALOX15 (leukocyte-type 12-LOX), belong to different enzyme families, although they exhibit a similar reaction specificity with AA as a substrate. The complex multiplicity of mammalian ALOX isoforms and the controversial enzyme nomenclatures are highly confusing and prompted us to summarize the current knowledge on the biological functions, enzymatic properties, and allosteric regulation mechanisms of mammalian ALOX15, ALOX15B, and ALOX12 orthologs that belong to three different enzyme sub-families.

## 1. Introduction

Mammalian arachidonate lipoxygenases (ALOXs) form a family of lipid-peroxidizing enzymes, which metabolize polyunsaturated fatty acids (PUFAs) to bioactive mediators commonly known as eicosanoids and related compounds [1,2,3]. Completion of the human genome project indicated the existence of six functional ALOX genes (*ALOX5*, *ALOX15*, *ALOX15B*, *ALOX12*, *ALOX12B*, *ALOXE3*), which encode for six different ALOX isozymes. In addition, several dysfunctional *ALOX* pseudogenes and a single *ALOX12* antisense gene (*ALOX12-AS1*) with unexplored biological relevance have been identified [4]. The corresponding ALOX isoenzymes are of biological relevance because of their involvement in different physiological processes and since they have been implicated in the pathogenesis of various diseases with great socio-economic impact [1,5].

Rabbit ALOX15 (rabbit reticulocyte 15-LOX1), which was discovered in 1975, has been comprehensively characterized with respect to its protein–chemical and enzymatic properties [6]. It was crystallized in 1997, and its 3D-structure was resolved at the atomic resolution of 2.4 Å [7,8]. Although the electron density map was incomplete, the X-ray coordinates of this enzyme served as the basis for structural homology modeling for other mammalian ALOX isoforms. More recent research provided the X-ray coordinates for a number of other mammalian ALOX variants: (i) ligand-free engineered human ALOX5 (PDB 3O8Y) [9], (ii) engineered human ALOX5 in complex with natural products (PDB 6NCF and 6N2W) [10], (iii) the catalytic domain inhibitor complex of pig ALOX15 (leukocyte-type 12-LOX, PDB 3RDE) [11], (iv) human ALOX15B (human 15-LOX2) with detergent (PDB 4NRE) [12], and v) human ALOX15B loop mutant in complex with an imidazole-based inhibitor (PDB 7LAF) [13]. In addition, several high-resolution cryo-EM structures of different oligomeric forms of human ALOX12 (platelet-type 12-LOX, according to the old nomenclature; PDB 8GHB to 8GHE) were recently reported [14].

According to the conventional lipoxygenase nomenclature, the different mammalian ALOX isoforms were named according to their positional specificity of arachidonic acid oxygenation. Following this concept, four different types of ALOX isoforms have been differentiated in various mammalian species (Figure 1): (i) AA 15-lipoxygenating enzymes, such as human and rabbit ALOX15 (15-LOX1) and human ALOX15B (15-LOX2); (ii) AA 12-lipoxygenating enzymes, such as the ALOX15 orthologs of mice, pigs, and cattle (leukocyte-type 12-LOX) as well as the ALOX12 orthologs (platelet-type 12-LOX) of all mammalian species (mammalian ALOX12B orthologs are also AA 12-lipoxygenating enzymes); (iii) AA 5-lipoxygenating enzymes, such as all mammalian ALOX5 orthologs (5-LOX); and (iv) AA 8-lipoxygenating enzymes, such as mouse Alox15b (mouse 8-LOX). 

Human individuals express two different types of AA 15-lipoxygenating ALOX isoforms, which were named ALOX15 and ALOX15B according to the gene-based enzyme nomenclature [15]. These two human ALOX isoforms share 38% of their amino acid sequence identity, and their 3D structures are very similar. However, the cellular distribution patterns and the biological functions of the two enzymes are very different, and thus, this structural multiplicity may not be considered a sign of functional redundancy [16]. Human ALOX15 produces a 10:1 mixture of 15(*S*)-HpETE [15(*S*)-hydroperoxyeicosatetraenoic acid, AA C-15 oxygenation] and 12(*S*)-HpETE [12(*S*)-hydroperoxyeicosatetraenoic acid, C12-oxygenation] as major AA oxygenation products. In contrast, human ALOX15B almost exclusively produces 15(*S*)-HpETE. It should, however, be stressed that the vast majority (>90%) of mammalian ALOX15 orthologs are AA 12-lipoxygenating enzymes [17,18]. Only mammals ranked in evolution above gibbons (*Hominidae*) and a limited number of other mammalian species, such as rabbits, mountain hares, anteaters, and bamboo rats, express AA 15-lipoxygenating ALOX15 orthologs [17,18]. Interestingly, mice do not express AA 15-lipoxygenating ALOX isoforms. In fact, the mouse Alox15 ortholog (leukocyte-type 12-LOX) dominantly forms 12-HETE from AA and mouse Alox15b (8-LOX) produces 8-HETE.

The main product of AA oxygenation by human ALOX12 (platelet-type 12-LOX) as well as of mouse, pig, and cattle ALOX15 (leukocyte-type 12-LOX) is 12(*S*)-HpETE. Chromosomal localization, sequence alignments, and comparison of the enzyme properties suggest that pig ALOX15 (leukocyte-type 12-LOX) and rabbit ALOX15 (reticulocyte-type 15-LOX1) belong to the same enzyme family despite their different reaction specificity with AA as a substrate. On the other hand, human ALOX12 (platelet-type 12-LOX) and pig ALOX15 (leukocyte-type 12-LOX) belong to different enzyme families although they exhibit a similar reaction specificity with AA as a substrate. In other words, the reaction specificity of an ALOX isoform is not a suitable parameter for ALOX classification since it overstresses a single enzyme property but completely ignores the evolutionary relations between different mammalian ALOX isoforms. For many years, mammalian ALOX isoforms have been believed to function as protein monomers with a single substrate-binding site. However, more recent data indicate allosteric properties of several ALOX isoforms, and these findings suggested more complex mechanisms for ALOX catalysis.

Among the human ALOX isoforms ALOX5 is of particular importance since it oxygenates AA to 5(*S*)-HpETE, which is further metabolized to pro-inflammatory leukotrienes. A PubMed search with the keyword “5-lipoxygenase” revealed some 6700 papers published over several decades. AA 15-lipoxygenases and AA 12-lipoxygenases attracted less attention among scientists; similar PubMed searches with the key words “15-lipoxygenase” (some 2200 hits) and “12-lipoxygenase” (1700 hits) revealed lower numbers of hits.

Despite the long history of ALOX research, the biological role of most mammalian ALOX isoforms is still a matter of discussion, and three major metabolic routes have previously been suggested [1,19,20,21,22,23,24,25]: (i) formation of bioactive mediators from free polyenoic fatty acids. Based on this concept, ALOX isoforms have been implicated in the biosynthesis of pro-inflammatory leukotrienes [19] as well as in the formation of anti-inflammatory and pro-resolving lipids [20]. (ii) Structural modification of complex lipid–protein assemblies such as biomembranes [21], lipoproteins [22], and epidermal ceramides [23]. According to this concept, ALOX isoforms may play a role in cell differentiation [5], atherogenesis [24], and ferroptosis [25]. (iii) Regulation of the cellular redox state, which is a key element of cellular genes expression. As pro-oxidative enzymes, ALOX isoforms oxygenate polyenoic fatty acids containing lipids to hydroperoxy derivatives, and thus, these enzymes upregulate the cellular oxidation potential. These alterations modify the activity of redox-sensitive transcription factors and, as final consequence, the expression levels of redox-sensitive genes [26]. Consequently, the functional phenotype of the cells is modified by the catalytic activity of these lipid-peroxidizing enzymes. In cellular systems, these three metabolic routes may even occur simultaneously and may interact with each other. Such interaction opens a wide range of potential biological functions for ALOX isoforms.

## 2. Discovery and Biological Function of Mammalian ALOX15, ALOX15B, and ALOX12 Orthologs

During the early days of ALOX research, it was widely believed that true lipoxygenase may not occur in animal cells [27]. However, in 1974, Hamberg and Samuelsson described the specific conversion of AA to 12-HETE by human blood platelets [28], and these data indicated for the first time that true ALOX isoforms may occur in mammals. This conclusion was later confirmed by the discovery of other ALOX isoforms [29], and today we know that the genomes of all mammals involve several functional ALOX genes [30].

Mammalian ALOX15 orthologs are highly expressed in reticulocytes. They have been implicated in cell differentiation [31,32,33], atherogenesis [34,35], ferroptosis [36,37], insulin resistance, and obesity [38,39,40]. These enzymes are capable of oxygenating not only free polyunsaturated fatty acids but also biomembranes and lipoproteins [41]. All mammalian ALOX15 orthologs exhibit a dual positional specificity, and they oxygenate free AA to variable amounts of 15(*S*)-HpETE and 12(*S*)-HpETE. The following 15(*S*)-HpETE: 12(*S*)-HpETE ratios were determined for different mammalian ALOX15 preparations: 95:5 for recombinant rabbit ALOX15, 92:8 for native rabbit ALOX15, and 90:10 for recombinant human ALOX15 [42,43,44]. ALOX15-derived metabolites of free polyenoic fatty acids (PUFAs) have previously been identified as PPAR-γ ligands with strong anti-proliferative activity [45]. The anti-inflammatory role of such metabolites has also been reported [46]. However, in different cancer types and in various inflammation models, ALOX15 and its PUFA metabolites exhibit dual functionality. For instance, ALOX15-derived linoleic acid (LA) metabolites may be pro-tumorigenic in glioblastoma cells [47]. 13(*S*)-HODE, the major ALOX15 metabolite of LA, induced suppression of PPARγ and stimulated tumor growth in prostate cancer via activation of the MAP kinase pathway [48]. The ALOX15 products 15-HETE and 13-HODE may play important roles in HCMV congenital infection [49].

Mouse Alox15 (leukocyte-type 12-LOX) is a close relative of human and rabbit ALOX15 (15-LOX1) [50], but its product pattern of AA oxygenation is different from those of human and rabbit ALOX15. In fact, mouse Alox15 oxygenates AA predominantly to 12(*S*)-HETE (90%), with 15(*S*)-HETE being a minor side product (10%). In vivo inactivation of the mouse *Alox15* gene allowed for investigations into the biological role of this enzyme in red blood cell development and in the pathogenesis of atherosclerosis.*Alox15*^−/−^ mice are characterized by reduced erythrocyte counts, elevated reticulocyte counts, and red cell hyperchromia. Erythrocytes of these animals were more susceptible to osmotic hemolysis and exhibited a reduced ex vivo life span. Transgenic expression of human ALOX15 in *Alox15*^−/−^ mice rescued the defective erythropoietic system and normalized the osmotic resistance. Taken together, these data suggest that mouse Alox15 plays a role in erythropoiesis, but the results also indicate that the lack of Alox15 is by no means lethal. An ALOX15 defect can be compensated by the upregulation of synergistic processes [51].

Human ALOX15 (15-LOX1) and its mouse ortholog (leukocyte-type 12-LOX) are capable of directly oxidizing polyenoic fatty acids incorporated into phospholipids and cholesterol esters present in lipoproteins. Since oxidation of lipoproteins has previously been implicated in the pathogenesis of atherosclerosis, several ALOX isoforms may play a role in this inflammatory vascular disease. *Alox15*^−/−^ mice were protected from aortic lipid deposition when the animals were fed a lipid-rich Western-type diet [16,52]. Post-ischemic heart function was significantly impaired in *Alox15*^−/−^ mice, and these data suggest a role of the enzyme in post-ischemic preconditioning [53]. Unfortunately, the molecular basis for the observed differences between *Alox15*^−/−^ mice and the corresponding wildtype control animals have not been explored in detail. Myocardium-specific knockout of the *Alox15* gene alleviated ischemia-reperfusion injury and restored cardiac function [54]. Here, 15(*S*)-HpETE, the minor side product of AA oxygenation by mouse Alox15, was identified as a trigger for cardiomyocyte ferroptosis [54]. Moreover, *Alox15*^−/−^ mice exhibited an altered lipidomic profile and different metabolic signatures than corresponding wildtype mice, and these differences were suggested to have delayed chronic heart failure progression and improved cardiomyocyte survival [55]. It should, however, be stressed that mouse Alox15 is an AA 12-lipoxygenating ALOX15 ortholog and thus, formation of 15-HpETE by this enzyme is limited. However, it might well be that the small amounts of 15-HpETE formed by mouse Alox15 may be sufficient to mediate the deleterious effect of Alox15 expression in this experimental system. If this is really the case, mice expressing the AA 15-lipoxygenating Leu353Phe mutant of Alox15 (the Phe353Ala mutant of mouse Alox15 is an AA 15-lipoxygenating enzyme) should be more sensitive to ischemia-reperfusion injury. Such *Alox15* knock-in mice are now available [56], and corresponding in vivo experiments can be carried out to confirm this interesting hypothesis.

In a cerebral stroke model, *Alox15*^−/−^ mice developed a reduced infarct area and less severe neurological dysfunction [57]. Here again, the formation of 15-HpETE by mouse Alox15 was suggested to play a major role. Since mouse Alox15 does only form small amounts of 15-HpETE, it remains unclear whether this minor AA metabolite is formed in sufficient amounts to mediate the deleterious effects of Alox15 in this stroke model.

In contrast to the deleterious roles of Alox15 in atherogenesis and stroke, a tissue-protective role of the enzyme in chronic inflammation was suggested. *Alox15*-deficient macrophages produce significantly reduced levels of lipoxin A_4_, which correlated with an activation of p38MAPK pathway and an enhanced expression of pro-inflammatory genes following stimulation with TNF-alpha. Unfortunately, the molecular basis for the cytoprotective role of the enzyme remains to be clarified [58]. Since AA 15-lipoxygenating ALOX15 orthologs exhibit an improved biosynthetic capacity for the formation of pro-resolving lipoxins [17], *Alox15 *knock-in mice expressing the Leu353Phe mutant [56] should be better protected from chronic inflammation than wildtype control animals.

Expression of ALOX12 (platelet-type 12-lipoxygenase) was reported in different types of human tumors, including prostate cancer, colorectal cancer, breast cancer, and lung cancer [59,60,61]. The enzyme is highly expressed in platelets [62] of different species, and the main AA oxygenation product of all mammalian ALOX12 orthologs tested so far is 12(*S*)-hydroperoxyeicosatetraenoic acid [12(*S*)-HpETE]. In other words, for this enzyme, there is no species-dependent reaction specificity, as is the case for mammalian ALOX15 and ALOX15B orthologs. Moreover, in contrast to mammalian ALOX15 orthologs, human ALOX12 (platelet-type 12-lipoxygenase) exhibits poor reactivity with PUFA molecules incorporated into the phospholipid bilayer of biomembranes. However, the formation of free 12(*S*)-HETE activated the NADPH oxidase pathway of neutrophils and macrophages, which led to an elevated formation of reactive oxygen species [63]. 12(*S*)-HETE has also been shown to function as ligand for the G-protein-coupled receptor 31 (GPR31), an orphan receptor initiating pro-thrombotic events [64]. Thus, inhibition of 12(*S*)-HETE formation is likely to impaired thrombus growth. 12(*S*)-HETE did also induce tissue factor-dependent thrombin generation following the re-esterification of this ALOX12 product into membrane phospholipids [65,66]. In *Alox12*^−/−^ mice, platelet aggregation is impaired [67], and taken together, these data suggest a regulatory role of this enzyme in blood clotting. Pancreatic islet cells of mice express both Alox12 (platelet-type 12-lipoxygenase) and Alox15 (leukocyte-type 12-LOX). Since the major AA oxygenation products of the two enzymes is 12(*S*)-HETE, it is difficult to puzzle out whether 12-HETE formation by these cells might be related to the expression of Alox15, Alox12, or both. Experiments with *Alox12*^−/− ^mice indicated that functional inactivation of the *Alox12* gene induces a compensatory upregulation of *Alox15* expression and sensitizes mouse pancreatic β cells for oxidative stress [68]. These data suggest that in mouse ß-cells, the Alox12 and the Alox15 pathway are obviously interconnected, but the details of this functional interactions have not been clarified on the molecular level.

Human ALOX15B (15-LOX2) was first described in 1997 as an AA 15-lipoxygenating enzyme that has a high level of expression in human skin, prostate, lung, and cornea [15]. Later on, the enzyme was also detected in human macrophages [69,70]. In contrast to human ALOX15 (15-LOX1), which forms a 9:1 mixture of 15(*S*)-HETE and 12(*S*)-HETE, human ALOX15B converts arachidonic acid almost exclusively to 15*(S*)-HpETE [15]. Although the biological function of ALOX15B remains a matter of discussion, the enzyme has been implicated in cell signaling [71]. In primary human macrophages, a regulatory role of ALOX15B in cholesterol homeostasis was proposed [72], and thus, the enzyme was linked to the pathogenesis of atherosclerosis [73]. Unfortunately, since no systemic *Alox15b*^−/−^ mice are currently available, the physiological and/or patho-physiological functions of Alox15b have not been tested in in vivo mouse atherosclerosis models. Since ALOX15B is expressed at high levels in human prostate and since enzyme expression is downregulated in prostate cancer, a tumor suppressor role of the enzyme has been suggested [74]. In fact, transgenic expression of human ALOX15B in mice under the control of a prostate-specific promoter induced cellular hyperplasia and cell senescence [74]. However, interpretation of these experimental data is somewhat problematic since mouse and human ALOX15B orthologs exhibit distinct catalytic activities. Human ALOX15B is an AA 15-lipoxygenating enzyme, but its mouse ortholog (mouse Alox15b, which was previously called mouse 8-LOX) forms 8(*S*)-HETE as the main AA oxygenation product. In other words, transgenic expression of human ALOX15B in mice is somewhat artificial since the transgenic enzyme produces different reaction products than the endogenous enzyme. A similar tumor-suppressor effect of human ALOX15B was suggested for breast cancer [75]. In contrast, expression of ALOX15B in colorectal cancer was associated with poor prognosis [76].

As for mammalian ALOX15 orthologs (see above), the differences in the catalytic properties of mouse and human ALOX15B orthologs (mouse Alox15b is an AA 8-lipoxygenating enzyme; human ALOX15B is an AA 15-lipoxygenating enzyme) need to be considered when scientists are working with mouse models of human diseases. Recently, *Alox15b* knock-in mice (*Alox15b*-KI) were generated [77] that express the AA 15-lipoxygenating Tyr603Asp+His604Val double mutant of mouse Alox15b instead of the AA 8-lipoxygenating wildtype enzyme. Although the catalytic activity of the recombinant Tyr603Asp+His604Val double mutant of mouse Alox15b exhibits reduced catalytic activity compared with the wildtype enzyme, the in vivo activity of wildtype and mutant enzyme species in PMA-treated tail skin was similar. These knock-in mice are fertile and reproduce like wildtype control animals. They display slightly modified plasma oxylipidomes but develop normally up to the age of 24 weeks. At later developmental stages, male *Alox15b*-KI mice gain significantly less body weight than wildtype controls. This effect is gender specific and might be related to the observation that male *Alox15b*-KI mice but not female individuals carry a dysfunctional erythropoietic system [77]. Moreover, *Alox15b*-KI mice are protected in the Freund complete adjuvant-induced paw edema inflammation model but not in the dextran sodium sulfate (DSS)-induced experimental colitis model [78].

## 3. Overall Structures of Mammalian ALOX15, ALOX15B, and ALOX12 Orthologs and Their Conformational Heterogeneity

### 3.1. Crystal and Solution Structure of Rabbit ALOX15 (15-LOX1)

The refined crystal structure of a rabbit ALOX15 inhibitor complex (resolution: 2.40 Å) [8] indicates two protein molecules in the asymmetric unit, and each protein involves 662 amino acids. Both monomers consist of two domains. The N-terminal PLAT (polycystin-1-lipoxygenase α-toxin) domain consists of eight parallel and anti-parallel β-sheets. The C-terminal catalytic domain, which involves the catalytic non-heme iron, involves 20 α-helices, 10 small 3_10_-helices, and 9 β-sheets in one monomer. The other monomer consists of 19 α-helices, 10 small 3_10_-helices, and 6 β-sheets. In the dimer structure (PDB 2P0M), one monomer (monomer B) carries the inhibitor [3-(2-octylphenyl)propanoic acid (RS7)] within the putative substrate-binding pocket. In contrast, the other monomer, monomer A, is not liganded. Monomer B, which carries the RS7 inhibitor, is present in a closed conformation, in which the access to the putative substrate-binding pocket is blocked by the orientation of the α2 helix. The ligand-free monomer A is present in an open conformation, in which the entrance to the substrate-binding pocket is freely accessible. However, the pocket is shallower than that of monomer B, and the α2 helix extends along nearly the entire length of the catalytic domain [8]. In the dimer, the two monomers are oriented towards each other by a pseudo-twofold axis that coincides with the crystallographic twofold axis [8]. The inter-monomer interface (Figure 2Aa), which is mainly formed by the two α2 helices, is stabilized mainly by hydrophobic interactions, but we detected hardly any hydrogen bonds. Except for the α2 and α18 helices, the spatial arrangement of the other secondary structural elements within the two monomers A and B is very similar (Figure 2Ab). In the liganded monomer B, the α18 helix as a whole is slightly dislocated compared with its counterpart in the ligand-free monomer A. On the other hand, the α2 helix in the liganded monomer B is strongly displaced by about 12 Å compared with the non-liganded monomer A (Figure 2Ab) [8].

As suggested by small-angle x-ray scattering (SAXS), rabbit ALOX15 in aqueous solutions may undergo reversible dimerization, and the relative shares of monomeric and dimeric enzymes depend on the conditions of the protein solution (Figure 2Ad) [80]. In contrast to human ALOX5, which forms covalently linked dimers [81], there is no covalent linkage between the monomers in rabbit ALOX15. The degree of rabbit ALOX15 dimerization depends on protein concentration, temperature, pH, ionic strength, and the presence or absence of active site ligands [79,80]. In low-ionic-strength buffers (20 mM Tris pH 6.8 to 8.0), a linear correlation between the monomer–dimer ratio and the enzyme concentration was observed [80], but under hypertonic conditions (200 mM NaCl), an increased degree of protein dimerization was detected. Purified recombinant rabbit ALOX15 migrated as a single band in native PAGE when the enzyme was kept in 20 mM Tris-HCl buffer containing 130 mM NaCl (near physiological conditions). In the absence of salt (20 mM Tris-HCl), two different protein bands were observed (Figure 2Ac) [82]. It is tempting to speculate that these two protein bands in native PAGE might represent the two enzyme conformers, but experimental proof of this assumption has not been provided. A similar doublet of protein bands was observed for the truncation mutant of rabbit ALOX15, which lack the N-terminal PLAT domain (Figure 2Ac) [82]. Thus, the differences in the electrophoretic properties of the two conformers may be related to a different surface charge, to a different overall shape, or to a mixture of both. However, inter-domain interactions might not be involved. Additional mutagenesis studies combined with size exclusion chromatography and electrophoretic mobility assays [82] confirmed that the conformational heterogeneity is an intrinsic property of the ALOX15 catalytic domain.

### 3.2. Structures of Human ALOX12 (Platelet-Type 12-Lox)

Unlike rabbit ALOX15, which occurs in a monomer–dimer equilibrium in aqueous solutions, high-resolution cryo-EM data on human ALOX12 (platelet-type 12-LOX) resolved different oligomeric states of the enzyme, ranging from enzyme monomers to enzyme hexamers. The structural architecture of monomeric ALOX12 is typical for lipoxygenases with an N-terminal PLAT domain and a mostly α-helical catalytic domain. Structural alignments with other ALOX structures revealed a similar fold of the enzyme [14]. Except for the ALOX12 dimer, the structures of the ALOX12 monomers in all oligomeric forms are similar. The individual monomers in the ALOX12 dimer have different structures, but neither of the monomers involved a ligand. In the ALOX12 dimer, the monomers are arranged “head to toe,” and most of the inter-monomer contacts are mediated via hydrophobic interactions between the α2 helices (Figure 2Ba). The ALOX12 monomers, which share a 60% amino acid identity with rabbit ALOX15, adopt either an “open” or a “closed” conformation. The structural difference between these two conformations is mainly related to the motion of the α2 helix and to corresponding rearrangements of the neighboring loops (Figure 2Bb). In addition, the structural orientation of the α2 helices in the “open” and in the “closed” conformations of human ALOX12 is different (Figure 2Bb). In the “open” conformation, the α2 helix forms a single helix that does not limit the entry of active site ligands into the putative substrate-binding pocket. Conversely, in the “closed” conformation, the α2 helix undergoes a 23° rotation, blocking the entrance to the putative substrate-binding pocket and reducing the volume of the pocket. The structural alteration of the α2 helix also leads to a 30° rotation of the two monomers relative to each other. Taken together, these structural data suggest that human ALOX12 shows a similar conformational heterogeneity as rabbit ALOX15, although the molecular events causing this heterogeneity are somewhat different for the two enzymes. In contrast to rabbit ALOX15, which exists as a mixture of protein monomers and protein dimers in aqueous solutions, SAXS analysis of human ALOX12 indicated that this enzyme is present in water as a mixture of oligomers, which tent to aggregate to higher-molecular-weight complexes [80,83].

### 3.3. Crystal Structure of the Catalytic Domain of Pig ALOX15 (Leukocyte-Type 12-Lox)

Since only low-quality crystals have been obtained for both wildtype full-length pig ALOX15 (leukocyte-type 12-LOX) and for several enzyme mutants, our knowledge on the 3D-structure of pig ALOX15 is based on X-ray diffraction data obtained for the catalytically active His-tag fused C-terminal domain (residues 112–663) of the enzyme in the presence of the inhibitor (4-(2-oxapentadeca-4-yne)phenylpropanoic acid; OPP). The protein crystals diffract at atomic resolution of 1.89 Å [11], and the overall structure of the truncated enzyme is similar to that of the catalytic domain of rabbit ALOX15. In the crystals, the truncated enzyme (PDB 3RDE) is present as a protein tetramer (Figure 2Ca), and the space symmetry group P 1 21 1 was observed. Each monomer carries an OPP molecule inside the putative substrate-binding pocket. In the tetramer structure, the individual monomers are linked via hydrogen bonds and salt bridges. Hydrophobic interactions are rare and the α2 helices of the monomers do not interact with each other, as is the case for rabbit ALOX15 (15-LOX1) and human ALOX12 (platelet-type 12-LOX). The overall structure of the catalytic domain of pig ALOX15 (Figure 2Cb) does align well with the ligand-free monomer A of rabbit ALOX15 (Figure 2Cc).

### 3.4. Crystal Structure of Human ALOX15B (15-LOX2)

The crystal structure of human ALOX15B (15-LOX2) was resolved at a 2.63 Å resolution. Although PDB 4NRE data only involve the X-ray coordinates for a single polypeptide chain, structural modeling using the PISA software suggested a protein hexamer composed of structurally identical monomers in H 3 2 symmetry. Each monomer involves 696 amino acid residues and contains two molecules of detergent (tetraethylene glycol monooctyl ether). One detergent molecule was located inside the protein and the other one on its surface. In the hexamer structure, the individual monomers are linked via hydrogen bonds and salt bridges, as well as by the detergent molecules. In contrast to rabbit ALOX15 (15-LOX1) and human ALOX12 (platelet-type 12-LOX), the α2 helixes of the different monomers may not play a major role in protein oligomerization. As expected, the human ALOX15B monomer displays the typical ALOX fold since each protein monomer involves an N-terminal PLAT domain and a helical catalytic subunit (Figure 2Da). Recently, the crystal structure of an ALOX15B loop mutant (deletion of residues 73–79) in complex with an imidazole-based inhibitor (PDB 7LAF) was resolved at a resolution of 2.44 Å. The complex represents a protein dimer that consists of two almost-equivalent 3-[(4-methylphenyl)methyl]sulfanyl-1-phenyl-1H-1,2,4-triazole-bound monomers (Figure 2Ea) in C 1 2 1 symmetry.

Human ALOX15B shares 40% of its amino acid sequence identity with rabbit ALOX15, and the overall 3D structures of the two enzymes are quite similar (Figure 2Db). However, there is an interesting structural difference, which might contribute to the different catalytic properties of the two enzymes. The α2 helix of human ALOX15B is shorter than that of rabbit ALOX15 and is somewhat rotated (Figure 2Db). Moreover, an overlay of the crystal structures of both monomers in the ALOX15B loop mutant dimer (PDB 7LAF) indicates subtle structural differences around the α2 helix (Figure 2Eb).

Like rabbit ALOX15, purified recombinant human ALOX15B migrated as a double band in native PAGE when the enzyme was kept in 20 mM Tris-HCl buffer without salt (Figure 2Da, inset) [82]. If these two protein bands represent two structurally distinct conformers, the enzyme as a whole might also exhibit a conformational heterogeneity. Unfortunately, there are currently no direct structural data available for this enzyme to support this hypothesis.

## 4. Inter-Monomer Interaction in Mammalian Alox Isoforms

### 4.1. Inter-Monomer Interaction in Mammalian ALOX Isoforms

Rabbit ALOX15 (15-LOX1) has a strong tendency for dimer formation, and the solvation energy of dimerization is about –25 kcal/mol. The area of the inter-monomer interface comprises 1335.0 Å^2^, and 37 amino acids of monomer A and 33 amino acids of monomer B contribute to this interface (Figure 3B). In the dimeric complex, two structurally different monomers A and B interact with each other mainly via their α2 helices [8]. In addition, an interaction between the α2 helix of one monomer and the α18 helices of the other monomer were observed in the crystal structure. Quantifying the relative contributions of individual amino acid residues to the inter-monomer contact plane using the PISA program [84] (www.ebi.ac.uk/msd-srv/prot_int/cgi-bin/piserver, accessed on 20 March 2024), we observed strong contributions of Leu183, Leu188, Leu192, Trp181, and His585. Site-directed mutagenesis and SAXS studies confirmed these modeling studies [79]. In fact, analysis of SAXS patterns suggested that the double-mutation Trp181Glu+His585Glu severely disturbed the dimerization of rabbit ALOX15 in solution [79].

Since no X-ray data are currently available for human ALOX15, the crystal structure of rabbit ALOX15 was used as a suitable template for homology modeling. Multiple amino acid sequence alignments of different mammalian ALOX15 orthologs indicated that Leu179, Leu183, and Leu192 of rabbit ALOX15 are conserved in human and pig ALOX15. Leu188 of rabbit ALOX15 aligns with Ile or Val residues in human (15-LOX1) and pig ALOX15 (leukocyte-type 12-LOX), and thus, the hydrophobic character of this residue is conserved (Figure 3A). Moreover, superposition of the 3D structures of the catalytic domains of the rabbit ALOX15 (monomer A) and the pig ALOX15 monomer (Figure 3C) indicate a similar position of the two α2 helices. Considering the high degrees of amino acid identity of rabbit ALOX15 on the one hand and of human and pig ALOX15 on the other (Figure 3A), it is highly probably that human and pig ALOX15 orthologs form similar assemblies as rabbit ALOX15.

The inter-monomer contacts in the human ALOX12 dimer appear to be somewhat weaker than those in rabbit ALOX15 since a lower solvation free energy gain of –19 kcal/mol was calculated. The inter-monomer interface of this enzyme (1051 Å^2^) is almost one third smaller than that of the rabbit ALOX15 dimer and involves 24 and 28 amino acid residues from the two monomers. (Figure 3D). For both enzymes, the monomers are linked mainly via their α2 helices. In addition, Trp595 of the α18 helix strongly contributes to the inter-monomer interactions. Sequence alignments of rabbit ALOX15 with human ALOX12 indicated that Leu188 of rabbit ALOX15 is conserved in human ALOX12. Other hydrophobic residues of the inter-monomer interface of rabbit ALOX15 are not conserved, and these data suggest different molecular forces for dimerization of the two enzymes. Leu183 and Leu194, which are located immediately downstream of the last turn of the α2 helix, largely contribute to the hydrophobic interactions that interconnect the two monomers of human ALOX12 (Figure 3D). Hydrogen bonds between Tyr191^A^/Ile206^B^ and Trp208^A^/Trp208^B^ may also contribute.

For human ALOX12, Leu183 and Leu187 play important roles for enzyme dimerization since Leu183Glu+Leu187Glu exchanges shift the oligomerization state of the enzyme from dimers to soluble monomers [85]. These data were further supported by comparison of wildtype human ALOX12 with its monomeric mutant Leu183Glu+Leu187Glu using hydrogen–deuterium exchange mass spectrometry (HDX-MS) [85]. This method is a powerful tool for testing protein conformational dynamics and protein–protein interactions. HDX-MS data of human ALOX12 indicated that the most dramatic effects were observed around the sites of Leu187Glu and Leu183Glu exchange.

Wildtype human ALOX15B monomers have similar overall structures as rabbit ALOX15 monomer A. In the dimeric crystal structure of the human ALOX15B loop mutant (PDB 7LAF), both monomers are linked in a “head-to-toe” way via their α2 helices. For these dimers (PDB 7LAF), the inter-monomer binding forces are even weaker than those of human ALOX12, with a solvation free energy of –12 kcal/mol. The inter-monomer interface of the ALOX15B loop mutant dimer is more than one third smaller than that of the rabbit ALOX15 dimer and involves 22 amino acids of monomer A and 24 residues of monomer B. The inter-monomer interface only comprises 808 Å^2^ (Figure 3E). The α2 helix of human ALOX15B (amino acids 178–197) is somewhat shorter than the corresponding helix of monomer A of rabbit ALOX15 (Figure 3E). It does not contain the Leu residues that have been implicated in the stability of the dimers of rabbit ALOX15 and human ALOX12 (Figure 3B,D). Instead, in human ALOX15B, Tyr185 and Ile197 mainly contribute to the inter-monomer interactions. In addition, salt bridges between Glu168 of monomer B and Arg203 or Arg215 of monomer A stabilize the dimer. In aqueous solution, ligand-free human ALOX15B is present as soluble monomer. This conclusion is based on data of hydrogen–deuterium exchange mass spectrometry (HDX-MS), which indicated deuterium incorporation into the peptide 185–192 (α2 helix). This high degree of deuterium incorporation suggests the accessibility of this structural element to solvent molecules [13,86].

### 4.2. Inter-Domain Interactions in Mammalian ALOX Monomers

All mammalian ALOXs constitute single polypeptide chain proteins that fold into a two-domain structure [87]. The PLAT domain comprises 100–110 amino acids. In the crystal structure of rabbit ALOX15 (15-LOX1), human ALOX12 (platelet-type 12-LOX), and human ALOX15B (15-LOX2), the two domains are tightly associated (Figure 4A–C).

For rabbit ALOX15 and human ALOX12, small-angle X-ray scattering [80,88,89] and membrane-binding assays [88] suggested that in aqueous solutions their PLAT domains may move relative to the catalytic subunit [80]. The degree of this inter-domain movement varies between ALOX isoforms and experimental conditions (buffer composition, pH, ionic strength, etc.). HDX-MS was recently employed to characterize the degree of dynamics of the two structural subunits in human ALOX15B in aqueous solution. The deuteration levels of tryptic peptides located in the PLAT domain and in the catalytic domain after H-to-D exchange are quite different [86]. For instance, the peptide 117–134, which represents the linker peptide that connects the PLAT domain with the catalytic subunit, experienced a high degree of H-to-D exchange, indicating its permanent contact with the solvent. In contrast, peptide 92–104, which contributes to the inter-domain interface, only shows moderate deuterium incorporation levels. These data suggest a limited water accessibility of this peptide. Unfortunately, no comparative experiments have been carried out with rabbit ALOX15 under identical experimental conditions, and thus, the degree of inter-domain movement in the two enzymes cannot directly be compared.

PISA evaluation (www.ebi.ac.uk/msd-srv/prot_int; accessed on 20 March 2024) of the inter-domain interface of rabbit ALOX15 suggested that the two domains share a joint average area of about 966 Å^2^. The area involves 29 amino acids of the catalytic domain and 26 residues of the N-terminal PLAT domain that are involved in hydrophobic interaction and form numerous hydrogen bridges (Figure 4A). The inter-domain interactions of human ALOX12 and human ALOX15B also involve hydrophobic interactions and hydrogen bonds, but for these enzymes the inter-domain contact planes are somewhat larger (1014 Å^2^ and 1104 Å^2^, respectively) (Figure 4B,C).

Tyr98 of the PLAT domain of rabbit ALOX15 significantly contributes to the interdomain contact plane (80 Å^2^). Size-exclusion chromatography and dynamic fluorescence studies on rabbit ALOX15 suggested a high degree of mobility of the N-terminal PLAT-domain relative to the catalytic subunit in mutants lacking aromatic side chains at position 98 [44]. These data suggest that in addition to hydrophobic interactions and hydrogen bridges, π–π interaction of Tyr98 (PLAT domain) and Tyr614 (catalytic domain) may also stabilize the inter-domain interactions (Figure 4A). Homology study indicates that Tyr98 in rabbit ALOX15 is conserved in human ALOX12 (Tyr97) and in human ALOX15B (Tyr107), whereas its counterpart Tyr614 is not conserved in human ALOX15B. Instead, an aromatic His is present at this position. Interestingly, Tyr98 and Y614 of rabbit ALOX15 are almost perfectly superimposed with the corresponding residues of both human ALOX12 and ALOX15B (Figure 4C,D, insets). Although Tyr107 in ALOX15B does not form any hydrogen bond with the corresponding residue of the catalytic domain, both Tyr98 and Tyr614 (rabbit ALOX15 nomenclature) may represent common anchoring groups that are involved in the inter-domain interaction of mammalian ALOX isoforms.

Another interesting point in the structural organization of rabbit ALOX15 is that Ser13, Ile14, and Tyr15 of the PLAT domain of rabbit ALOX15 (PDB 2P0M) interact with α2 helix residues (Leu168, Glu169, Asp170) of the catalytic subunit and, thus, may contribute to the inter-domain interactions (Figure 4A). When the N-terminal domain swings away from the catalytic subunit, the beginning of this helix becomes destabilized and may lose its secondary structure at elevated temperatures. Indeed, mutation of Ile14 to a positively charged Lys increased the thermal stability of the protein by forming an additional salt bridge between the PLAT domain (Lys14) and the initial part of the α2 helix (Asp170) of the catalytic domain. Analysis of the temperature-induced denaturation of wildtype ALOX15 and its Ile14Lys mutant at 37 °C suggested that, in contrast to the Ile14Lys mutant, the wildtype enzyme loses a segment of its α-helical structure, which comprises 25 amino acids.

In contrast to the structure of rabbit ALOX15, no direct interactions of residues of the PLAT domain with those of the α2 helix were observed in human ALOX12 or in human ALOX15B. However, hydrogen bonds were detected between Trp99 or Trp109 and Asn163 or Asn173 of ALOX12 and ALOX15B, respectively. Both Asn163 (human ALOX12) and Asn173 (human ALOX15B) belong to small 3_10_ helices that are located immediately upstream of the α2 helices of the two enzymes. Unfortunately, it is not clear whether these interactions may contribute to enzyme stability.

## 5. Mechanism of ALOX Catalysis and Mechanistic Basis for the Positional Specificity of ALOX Isoforms

### 5.1. General Mechanism of ALOX Reaction

The putative substrate-binding pockets in the catalytic domain of all mammalian ALOX isoforms harbor a catalytically active non-heme iron. In rabbit ALOX15 (15-LOX1), the iron ion is coordinated by five amino acid side chains (His361, His366, His541, His545, C-terminal Ile). A water molecule or an OH^−^ ion was identified as the sixth iron ligand [90]. In human ALOX12 (platelet-type 12-LOX), the catalytic iron is coordinated by three conserved His residues (His360, His365, His540) by Asn544 and the carboxylic group of the C-terminal Ile663 [14]. Interestingly, in human ALOX15B, the iron is liganded only by four amino acid side chains (His373, His378, His553, C terminal Ile676) and by two water molecules [12]. Ser557, which aligns with the iron ligand His545 of rabbit ALOX15, is not an immediate iron ligand in human ALOX15B.

During ALOX catalysis, the iron ion shuttles between its ferrous (Fe^2+^) and ferric (Fe^3+^) states. According to the generally accepted mechanism, the ALOX reaction consists of four elementary reactions (hydrogen abstraction, radical rearrangement, oxygen insertion, peroxy radical reduction), and in most cases, the stereochemistry of these elementary reactions is tightly controlled by the enzyme (Figure 5). The initial and rate-limiting step of ALOX-catalyzed PUFA oxygenation is activation of a bisallylic methylene group (Figure 5) [91]. This activation proceeds as proton-coupled electron transfer (PCET), in which the proton is transferred to the Fe^3+^-bound hydroxide, forming water, and the electron is accepted by the oxidized iron, reducing it to its ferrous form (Fe^2+^) [92]. The transferred electron is not localized on the proton but tunnels directly from the substrate to the ferric iron in a concerted proton-tunneling–electron-tunneling process [93]. The initial activation step is followed by radical rearrangement so that the electron density of the carbon-centered fatty acid radical is focused on either C1 or C4 of the pentadiene system. In the transition state, the Fe-O-H-C bridge lowers the energy barrier for hydrogen abstraction and provides an efficient pathway for coupled electron–hydrogen tunneling.

Under normoxic conditions, oxygen insertion into the delocalized fatty acid radical proceeds in a regio- and stereo-selective manner, forming oxygenated PUFAs in either the (*S*)-(ALOX15, ALOX15B, ALOX12, ALOX5) or the (*R*)-(ALOX12B) configuration [2,94]. Rearrangement of the carbon-centered fatty acid radical is followed by the addition of molecular dioxygen at either the *n* − 2 carbon or the *n* + 2 carbon of the fatty acid (Figure 5), and this oxygen insertion proceeds antarafacially (on the opposite face of the plane, determined by the two double bonds of the pentadienoic system) to hydrogen abstraction.

### 5.2. Structural Basis for the Reaction Specificity of Mammalian ALOX15 Orthologs

In its ligand-free form (monomer A), the putative substrate-binding cavity of rabbit ALOX15 is a funnel-shaped cavity that reaches the protein surface at Arg403. In its ligand-bound form (monomer B), the substrate-binding pocket is deeper and concaved by the side chain of Leu408. The walls of the cavity are formed by 23 predominantly hydrophobic amino acids, which are localized in six different helices (H2, H7, H9, H10, H16, H18) and by the loop connecting H9 and H10. The entrance to the putative substrate-binding pocket in the ligand-bound monomer B appears to be closed. In accordance with previous observations, Arg403 might be involved in fatty acid binding [95]. The interaction of the substrates’ carboxylic group with the positively charged Arg403 rearranges its original contacts and might trigger conformational alterations around the α2 helix, favoring the formation of the catalytically competent ALOX15 dimers [96,97].

The X-ray coordinates of rabbit ALOX15 and multiple amino acid sequence alignments of mammalian AA 12- and AA 15-lipoxygenating ALOX15 orthologs suggest that the amino acids at the positions Phe353, Ile418/Met419, and Ile593 form the bottom of the substrate-binding pocket. The geometry of their side chains determines how deep a fatty acid substrate may slide into the active site [98,99,100,101]. Alanine-scan mutagenesis studies of the four specificity determinants (Phe353, Ile418, Met419, Ile593) suggested that Ile418 and Phe353 most strongly contribute to the positional specificity of rabbit ALOX15. On the basis of additional mutagenesis data, three regions of the primary structure of mammalian ALOX15 orthologs are important for the positional specificity of these enzymes: (i) the region around Ile418 and Met419 (Sloane determinant) [98,99], (ii) the region around Phe353 (Borngräber 1 determinant [100], and (iii) the region around Ile593 (Borngräber 2 determinant) [101]. Site-directed mutagenesis studies on the ALOX15 orthologs of humans [100], rabbits [101], pigs [102], and a large number of other mammals [103,104] support the triad concept (Figure 6).

The relative importance of the triad determinants varies for different mammalian ALOX15 orthologs. In pig ALOX15, Ile418 and Met419 of human and rabbit ALOX15 (numbering according to rabbit ALOX15) are present as less space-filling Val residues, and Ile418Val+Met419Val exchange in rabbit ALOX15 converted the AA 15-lipoxygenating wildtype enzyme to an AA 12-lipoxygenating enzyme species (Figure 6). As with most mammals, mice express an AA 12-lipoxygeting Alox15 ortholog. Interestingly, for this enzyme, the Sloane determinants are less important for the reaction specificity. Here, the Borngraber-1 determinant plays a major role. In mouse Alox15 and in the ALOX15 orthologs of a large number of other mammals [18], the Borngraber-1 determinant is occupied by either Leu or Ile residues, which are less space-filling than Phe353, which is present at this position in human and rabbit ALOX15. Phe353Ala exchange in human ALOX15 completely murinized the reaction specificity of human ALOX15, converting the AA 15-lipoxygenating enzyme to an AA 12-lipoxygenating protein. An inverse mutagenesis strategy on mouse Alox15 humanized the reaction specificity of the recombinant enzyme. Recently, *Alox15* knock-in mice were generated, which express the AA 15-lipoxygenating Leu353Phe mutant of mouse Alox15 instead of the AA 12-lipoxygenating wildtype enzyme [56]. These data indicate that the reaction specificity of human and mouse ALOX15 orthologs can easily be inter-converted by a single point mutation and that the triad concept does also work in vivo. Interestingly, when we checked different human SNP databases to explore whether a Phe353Ala mutation in the human *ALOX15* gene might occur in human individuals, we did not find a single hit. These data suggest that such a mutation might be prevented by evolutionary pressure, but it remains unclear what this pressure might actually be.

### 5.3. Structural Basis for the Reaction Specificity of Human ALOX12

The cryo-EM structures of the human ALOX12 tetramer (1.7 Å) and the corresponding hexamer (2.3 Å) were employed to model AA as substrate in the electron density map of this enzyme [14]. In these structures, the entrance to the putative substrate-binding pocket is limited by arched helices (helices 14–15) and the extended α2 helix. The carboxyl group of AA may interact with His596 [105], positioning the bisallylic system around C10 of the substrate molecule acid in the vicinity of the catalytic iron. The active site of human ALOX12 is a U-shaped cavity that is also lined by hydrophobic residues. Among the amino acids surrounding the fatty acid tail are hydrophobic residues including the triad determinants (Phe352, Val418, Ile593). It should, however, been stressed that the triad concept is only partly applicable for human ALOX12 [103]. In fact, corresponding amino acid exchanges did not induce complete inversion of the reaction specificity or lead to inactive enzyme mutants.

The invariant Leu407 of the arched helix (helices 14–15) is located in the hinge region of the U-shaped substrate-binding channel. Mutation of this residue to Ala enlarged the volume of the putative substrate-binding cavity and strongly (100-fold) reduced the first- and second-order rate constants. Interestingly, the pattern of the AA oxygenation products was hardly altered [105]. Leu 597 is located across the substrate-binding pocket opposite Leu 407, and mutagenesis of this residue should induce similar changes in the catalytic properties. Unfortunately, mutagenesis studies on this amino acid have not been published.

As suggested for the rabbit ALOX15 dimer, only one monomer of the ALOX12 dimer (Figure 2B) may be accessible for active site ligands. The substrate-binding pocket of the other monomers of the two enzymes might be blocked since the α2 helices might close the entrance.

### 5.4. Structural Basis for the Reaction Specificity of Human ALOX15B

The U-shaped substrate-binding pocket of human ALOX15B involves the triad determinants (Phe 365, Ile 412,), as well as Thr431 and Leu420, which contribute to properly aligning the reaction-relevant pentadiene system of AA (C11–C15) for hydrogen abstraction. There are small differences in the positions of these amino acids in the original wildtype human ALOX15B structure, which contain detergent molecules, and the new structure of a mutant ALOX15B inhibitor complex, but these differences may not dramatically impact fatty acid substrate binding [13].

The triad concept, which nicely explains the reaction specificity of all mammalian ALOX15 orthologs tested so far, does not explain the reaction specificity of mammalian ALOX15B orthologs [103]. In fact, for these enzymes, other amino acids (Jisaka determinants) may be much more important. Human ALOX15B is an AA 15(*S*)-lipoxygenating enzyme [15], but the major AA oxygenation product of mouse Alox15b (8-LOX) is 8(*S*)-HETE [106,107]. Mutagenesis of the Jisaka determinants of mouse Alox15b ((Tyr603, His604) to the corresponding amino acids present at these positions in human ALOX15B (Asp602, Val603) completely humanized the reaction specificity of mouse Alox15b [106]. The inverse mutagenesis strategy on human ALOX15B partly murinized the reaction specificity of this enzyme, with 8(*S*)-HETE being the dominant AA oxygenation product [106]. These mutagenesis data can be explained when an inverse head-to-tail substrate orientation is assumed for human and mouse ALOX15B orthologs (Figure 7). This inverse substrate orientation might be forced by the positive charge introduced by the Val-to-His exchange. To test whether mutation of the Jisaka determinants does also modify the reaction specificity of mouse Alox15b in vivo and to explore whether a modification of the reaction specificity might be of biological relevance, *Alox15b*-knock-in mice (*Alox15b*-KI mice) were recently generated that express the AA 15-lipoxygenating Tyr603Asp+His604Val double mutant of mouse Alox15b instead of the AA 8(*S*)-lipoxygenating wildtype enzyme [77].

As indicated above, the reaction specificity of mammalian ALOX15 orthologs depends on the evolutionary ranking of the animals, and the evolutionary hypothesis of ALOX15 specificity [18] suggests that all mammals ranked in evolution above gibbons express AA 15-lipoxygenating ALOX15 orthologs. In contrast, most (<90%) mammals ranked in evolution below gibbons, including mice, rats, and pigs, as well as other primates, express AA 12-lipoxygenating enzyme orthologs. To explore whether there is a similar evolution dependence for the reaction specificity for mammalian ALOX15B orthologs, we recently retrieved the amino acid sequences of more than 90 mammalian ALOX15B orthologs from different public sequence databases and compared the sequence motifs of the Jisaka determinants. Here, we found that independent of the evolutionary ranking of the animals, these positions were occupied by four different amino acid motifs: Asp-Val, Asp-Ile, Asp-Met, and Tyr-His. Mutagenesis experiments on human ALOX15B indicated that the three enzyme mutants carrying the Asp-Val, Asp-Ile, and Asp-Met motifs converted AA almost exclusively to 15(*S*)-HETE. When we expressed 20 of these mammalian ALOX15B orthologs, including the enzymes of two different rat species (*R. rattus*, *R. norvegicus*), we confirmed that all of them were AA 15(*S*)-lipoxygenating enzymes. Only the ALOX15B orthologs of different species of the genus *Mus* (*M. musculus*, *M. Pahari*, *M. caroli*) and of the rice-field mouse (*Mastomys coucha*) carried the Tyr-His motif at the Jisaka positions, and the recombinant enzyme were identified as AA 8(*S*)-lipoxygenating ALOX1B5 orthologs. In summary, these data, which have not been published so far, indicate that the reaction specificity of mammalian ALOX15B orthologs does not depend on the evolutionary ranking of the mammals. The vast majority of mammalian ALOX15B orthologs are AA 15(*S*)-lipoxygenating enzymes, and the 8(*S*)-lipoxygenating activity of mouse Alox15b is an evolutionary exception. In other words, the difference in the reaction specificity of human ALOX15B and mouse Alox15b orthologs is not part of an evolutionary concept aimed at optimizing the functional properties of these enzymes, as has been suggested for mammalian ALOX15 orthologs [17,18].

### 5.5. Stereo-Control of Oxygen Insertion

As indicated above, several amino acids have been identified to control the regio-selectivity of different ALOX isoforms (AA 12-oxygenation vs. AA 15-lipoxygenation), but the molecular basis for the enantio-selectivity of most ALOX isoforms is still not completely understood. Most human ALOX isoforms, such as human ALOX15, human and mouse ALOX15B, human and mouse ALOX12, and human and mouse ALOX5, are (*S*)-lipoxygenases, which almost exclusively convert AA to 15(*S*)-HETE, 12(*S*)-HETE, 8(*S*)-HETE, and 5(*S*)-HETE. The corresponding (*R)*-enantiomers are only generated in small amounts. These enzymes are called (*S*)-lipoxygenases. On the other hand, human ALOX12B, which predominantly oxygenates AA derivatives to 12(*R*)-HETE, constitutes an AA (*R*)-lipoxygenase. Similar catalytic properties have been described for mouse Alox12b. The enantioselectivity of (*R*)- and (*S*)-lipoxygenases has been related to the orientation of the substrate fatty acids within the substrate-binding pocket [108,109], but the local availability of molecular dioxygen at the active site of the enzyme may also be of major importance. The dioxygen molecules, which are introduced into the fatty acid substrate during the ALOX reaction, must diffuse from the surrounding solvent into the substrate-binding cleft, and the path of intra-protein oxygen diffusion is important for the stereochemistry of oxygen insertion. If the oxygen concentration at a certain position in the substrate-binding pocket is higher than at other positions, molecular dioxygen should preferentially be inserted at this position into the carbon-centered fatty acid radical. In other words, oxygen availability within the substrate-binding pocket of ALOX isoforms may be of major relevance for the enantioselectivity of the enzymes. Unlike in water, oxygen is not uniformly distributed in proteins. In fact, there are regions with high and regions with low oxygen concentrations. Other regions might be completely devoid of oxygen. For a long time, it remained unclear how oxygen penetrates from the periphery of ALOX isoforms into the catalytic center. Choosing rabbit ALOX15 as a model system, Saam and co-workers [110] evaluated the oxygen distribution within the ALOX15 protein and defined potential routes for intra-protein oxygen diffusion. They first computed the 3D free-energy distribution for oxygen inside the ALOX15 protein, which led to the identification of four partly overlapping oxygen access channels. These channels connect the protein surface with the region of highest oxygen affinity, which is located at the active site. This region was localized opposite to the non-heme iron. Interestingly, the catalytically most relevant path of intra-protein oxygen diffusion was partly obstructed by Leu367Phe exchange, which led to a strongly increased Michaelis constant for oxygen for the mutant enzyme. The blocking mechanism was explained in detail by reordering the hydrogen-bond network of water molecules detected at this position of the active site. These modeling studies identified putative oxygen access channels in rabbit ALOX15 but did not explain the high degree of stereo-selectivity of the enzyme. In fact, the oxygen concentrations were similar at the pro*R* and the pro*S* positions of the fatty acid radical. In other words, the differences in the oxygen concentrations at the catalytic center of the enzyme are not high enough to explain the high degree of enantioselectivity of rabbit ALOX15. Thus, other mechanisms may play a more important role.

The Ala-vs-Gly hypothesis of ALOX specificity [109] suggested that a single amino acid residue in the catalytic domain of all ALOX isoforms is important for stereo control of the ALOX reaction (Figure 8). This residue is conserved as Ala in a number of (*S*)-lipoxygenases but is a Gly in several (*R*)-lipoxygenases. Ala-to-Gly mutations in two mammalian (*S*)-lipoxygenases (mouse Alox15b and humanALOX15B) and the inverse Gly-to-Ala substitution in two (*R*)-lipoxygenases (human ALOX12B and coral 8R-LOX) inverted the enantio-selectivity of the enzymes. The authors concluded that the presence of an Ala at this critical position favors oxygenation of the pentadiene radical at its distal end [n + 2], which leads to (*S*)-stereochemistry of the product hydroperoxide. In contrast, when a less bulky Gly residue is present at this position, the proximal end [n-2] of the pentadiene radical is oxygenated, which results in the formation of the (*R*)-hydroperoxide. Unfortunately, the A-vs-G concept is not applicable for all lipoxygenases [103], and zebrafish ALOX1, which carries a Gly at this position, converts AA almost exclusively to 12(*S*)-HETE [111,112]. This is also the case for the ALOX15B orthologs of other bony fish species [113].

Using a combination of quantum mechanics/molecular mechanics calculations with molecular dynamics simulations, Suardiaz and co-workers explored the mechanism of molecular dioxygen insertion into the AA-derived pentadienyl radical by rabbit ALOX15 [114]. They found that the 15-HETE/11-HETE (C15/C11) ratio for the Leu597Val mutant of this enzyme was ten times lower than that of the wildtype enzyme. However, the (*S*)-stereochemistry of the major products was kept. In contrast, the Leu597Ala favored oxygen insertion at C15 of the AA backbone, but (*R*)-stereochemistry was favored. As molecular basis for this *S*-to-*R* exchange, the authors discussed that the Leu-to-Ala exchange provided more space at the active site so the fatty acid substrate could adopt a different position within the substrate-binding pocket so that oxygen insertion at the 15(*R*)-position was favored. Unfortunately, these conclusions were drawn on the basis of in silico modeling studies only, and experimental confirmation on recombinant rabbit ALOX15 is still pending. Interestingly, as both the Ile593 (Borngräber 2 determinant) and Leu597 belong to the α18 helix, this helix may play a crucial role in positioning the substrate fatty acids in the substrate-binding cavity. Additionally, Gln596, which is invariant in numerous mammalian ALOX isoforms, has been proposed to fine-tune the reaction specificity, and depending on the fatty acid substrate’s structure, it may also interact with its carboxyl group [115].

## 6. ALOX Inhibitors

The potential patho-physiological roles of different ALOX isoforms in various types of human diseases made these enzymes promising targets for pharmacological research. Unfortunately, studies with different types of ALOX knockout mice sometimes provide controversial data, so the patho-physiological roles of the different ALOX isoforms in mouse models of human diseases are still a matter of discussion. Moreover, different ALOX isoforms biosynthesize similar reaction products, and thus, it is difficult to puzzle out which of the enzymes is responsible for the in vivo formation of a specific lipid mediator. These problems could at least partly be solved by the use of isoform- and/or ortholog-specific ALOX inhibitors. In the past, a number ALOX inhibitors have been developed, but some of the frequently employed compounds do not exhibit a high degree of isoform specificity [116]. Moreover, some of these compounds induce off-target effects, which leads to misinterpretations of experimental data.

Based on their mechanisms of action, four different groups of ALOX inhibitors can be separated. (i) Redox inhibitors: Redox inhibitors interfere with the valency change in the enzyme-bound non-heme iron and thus inhibit the ALOX reaction [117,118]. Since the valency change is similar in all ALOX isoforms, redox inhibitors might not exhibit a high degree of isoform specificity as long as they can penetrate the substrate-binding cavities of different ALOX isoforms. (ii) Competitive and non-competitive active site probes: Active site probes are bound in the substrate-binding cleft of ALOX isoforms and may prevent the binding of fatty acid substrates [119]. Although the overall 3D-structure of most ALOX isoforms is similar, there are subtle isoform-specific structural differences that can be used to design isoform-specific inhibitors in the frame of rational drug design. (iii) Suicide substrates: Acetylenic fatty acids such as eicosatetraynoic acid (ETYA) are bound at the active site of ALOX isoforms, and hydrogen abstraction is catalyzed. This reaction leads to the formation of reactive intermediates, which may rapidly interact with active site amino acids, leading to enzyme inactivation [120]. Unfortunately, suicide substrates may bind at the active site of most ALOX isoforms, and thus, they are likely to lack a high degree of isoform specificity. (iv) Allosteric inhibitors: These compounds bind either to a putative allosteric binding site of monomeric ALOX isoforms or at the active site of one ALOX monomer (allosteric monomer) within dimeric enzymes. In both cases, this binding may alter the structure of the enzyme and thus its catalytic properties [121].

The chemistry of ALOX inhibitors is rather complex, and a large number of lead compounds have been explored [122]. The chemical structure of selected ALOX inhibitors is presented in Table 1. Indole-based inhibitors (compound **1** in Table 1) [121] were identified in kinetic spectrophotometric assays, and these compounds exhibited high inhibitory potency for purified recombinant rabbit and human ALOX15. Unfortunately, the inhibitory potency for other human ALOX isoforms (ALOX15B, ALOX12, ALOX5, ALOX12B, ALOXE3) has not been tested using the same kinetic assay system.

The tryptamine derivative (compound **2** in Table 1) exhibited a high inhibitory potency for purified native rabbit ALOX15 [123], which was obtained from phenylhydrazine-treated rabbits. ALOX inhibition was tested in a standard colorimetric assay at low substrate concentrations [126]. Here, a 100-fold selectivity of compound **2** was observed for native rabbit ALOX15 over recombinant human ALOX5 and ALOX12 purchased from commercial sources (Cayman Chem). Unfortunately, the assay systems used in this study to test the catalytic activity of native rabbit ALOX15 on the one hand and of recombinant ALOX5 and ALOX12 on the other significantly differ from each other. These differences might lead to misinterpretation of the experimental data. Moreover, the quality of the enzyme preparations (e.g., iron content) was not well characterized, and the inhibitory potency for other ALOX isoforms (ALOX15B, ALOX12B, ALOXE3) has not been tested.

Another class of compounds is imidazole-based inhibitors (compounds **3** and **4** in Table 1) [124]. These compounds were highly potent for purified rabbit ALOX15, but no isoform selectivity data were provided. Different five-membered pyrazole-based sulfonamides (compound **5** in Table 1) exhibited a high inhibitory potency for purified rabbit ALOX15 [125], but here again, the inhibitory power for other ALOX isoforms has not been tested.

An oxazole derivative (ML351, compound **6** in Table 1, IC_50_ = 200 nM) was identified as a highly potent inhibitor of recombinant human ALOX15 [119] in a colorimetric high-throughput screening assay. In subsequent spectrophotometric assays, this compound exhibited a high specificity for human ALOX15 compared with purified recombinant human ALOX12 and crude recombinant human ALOX5 preparations. In these spectrophotometric assays, low (two-digit micromolar) AA concentrations were used and detergents were added to improve the substrate availability. Interestingly, when ML351 was tested in an assay system, in which the formation of ALOX products was quantified by HPLC, the isoform specificity could not be confirmed [116]. Here, human ALOX15, human ALOX5, and mouse Alox5 were inhibited at 10 µM inhibitor concentrations, whereas mouse Alox15, human ALOX12, mouse Alox12, human ALOX15B, and mouse Alox15b were not inhibited. In this assay system, AA was present at much higher concentrations (100 µM), no detergents were included, and crude recombinant enzyme preparations (bacterial lysate supernatants) were used. Although the reaction specificity of all enzyme preparations was checked as quality criteria, the specific catalytic activities and iron content of the enzymes were not quantified. If one compares the outcomes of these two inhibitory studies, one may conclude that ML351 exhibited a pronounced isoform specificity for human ALOX15 under certain assay conditions [119] but that this isoform specificity was much less pronounced under different experimental conditions [116]. The different substrate concentrations used in the two studies and the presence or absence of detergents might have contributed to the opposite conclusions.

As consequence of the conflicting results obtained for ML351 in different inhibitor studies, the following rules for future searches for isoform-specific ALOX inhibitors may be recommended: (i) The inhibitory potencies of a given compound should always be quantified on recombinant mouse and human enzyme preparations, preferentially using purified enzymes. The use of other enzyme preparations (crude cellular extracts) should be avoided. (ii) Similar fatty acid oxygenase activities of the different enzyme preparations should be employed, and these activities should be corrected for the iron content since this ensures high-quality enzyme preparations. (iii) Identical assay conditions (pH, T, buffer, kind of substrate, substrate concentration, presence and absence of detergents) should always be used for the different ALOX isoforms. (iv) The assay method (colorimetry, oxygraphy, spectrophotometry, HPLC product analysis) should be the same for all enzymes.

In the search for novel ALOX inhibitors, arachidonic acid is usually used as a substrate, but other polyenoic fatty acids have not been tested. Interestingly, for some ALOX15 inhibitors (compounds **1** and **4**, Table 1), different inhibitory potencies were reported for purified recombinant rabbit ALOX15 when AA and LA were used as substrates. In fact, the IC_50_-values of these compounds for AA and LA oxygenation did differ by several orders of magnitude. In other words, compounds **1** and **4** were effective inhibitors of ALOX15-catalyzed LA oxygenation but not of AA oxygenation catalyzed by the same enzyme in identical assay systems. These findings were quite surprising, and considering the fact that both substrates have comparable affinities for purified recombinant ALOX15, the differences in the IC_50_ values suggest that the mechanism of lipoxygenase catalysis appears to be more complex than anticipated and may involve allosteric regulatory elements.

An oxadiazole-based inhibitor, MLS000545091 [127] (compound **7** in Table 2), was identified using a spectrophotometric assay as a first potent and selective mixed-type inhibitor of purified human ALOX15B [128]. This compound exhibited 20-fold selectivity for human ALOX15B over purified human ALOX15 [129], purified human ALOX12 [130], and crude human ALOX5 preparations [131]. Later, a novel chemotype that contained a central imidazole ring substituted at the 1-position with a phenyl moiety and with a benzylthio moiety at the 2-position was found to be potent and selective against ALOX15B. Compound **8** [13] (Table 2) displayed at least 50-fold selectivity for purified human ALOX15B versus purified human recombinant ALOX15 and purified recombinant human ALOX12. Here again, isoform specificity was assayed at low substrate concentrations, and detergents were included to improve substrate availability. N-((8-hydroxy-5-nitroquinolin-7-yl)(thiophen-2-yl)methyl)propionamide (compound **9** in Table 2, NCTT-956), which was developed using the metal chelator 8-hydroxyquinoline (8-HQ) as the lead compound, displayed high inhibitory potency (IC_50_ of 800 nM) for purified recombinant human ALOX12. Its selectivity over purified recombinant human ALOX15 was higher than 25-fold [132]. Structural optimization of the 4-((2-hydroxy-3-methoxybenzyl)amino)benzenesulfonamide scaffold led to the development of compound **10** (ML355, Table 2), which constitutes a potent human ALOX12 inhibitor that displayed 30-fold selectivity over purified human ALOX15 and 100-fold selectivity over purified recombinant human ALOX15B and over crude human ALOX5 [133]. At higher substrate concentrations and in the absence of detergents, ML355 did also inhibit crude recombinant enzyme preparations of human ALOX15 and human ALOX5 [116]. Finally, the naphthyl-benzothiazole derivative of ML355 (Lox12Slug001, **11**) exhibited seven-fold higher inhibitory potency than ML355 and 100-fold higher selectivity over purified recombinant human ALOX15 [129] and ALOX15B [128], and crude preparations of human ALOX5 [131]. However, it remains unclear whether this pronounced isoform specificity can also be confirmed at higher substrate concentrations in the absence of detergents and whether the epidermal ALOX isoforms (ALOX12B and ALOXE3 in humans; Alox12b, Aloxe3, and Aloxe12 in mice) are also inhibited.

Human ALOX12 predominantly oxygenates AA as a substrate, whereas ALOX15B may utilize both LA and AA. Although recombinant human ALOX15B and mouse Alox15b strongly prefer AA (ten-fold higher oxygenation rate) over LA, the product pattern of human ALOX15B with LA is very specific, and these data indicate that the reaction proceeds to be enzyme controlled. Unfortunately, there are no reports in the literature on whether selective ALOX15B inhibitors have been tested with LA as a substrate and whether they may act in a substrate-specific manner, as observed for ALOX15.

Another important point that needs to be considered in the search for effective ALOX inhibitors is their ortholog specificity. Since mouse Alox15 (leukocyte-type 12-LOX) and human ALOX15 (15-LOX1) orthologs exhibit different catalytic properties, whether an effective inhibitor of human ALOX15 does also effectively inhibit mouse Alox15 always needs to be tested. A lack in ortholog specificity is very important for further drug development since many experiments in later stages of drug development must be carried out in mice. If a given compound inhibits human ALOX15 but does not inhibit mouse Alox15, it may not be meaningful to further develop this compound. This may also be the case for human ALOX15B and mouse Alox15b orthologs since these two enzymes also exhibit distinct catalytic properties. However, this is not the case for mouse and human ALOX12 and ALOX5 orthologs.

## 7. Allosteric Effectors in ALOX Catalysis

### 7.1. ALOX Isoforms Are Allosteric Enzymes

A number of ALOX isoforms, such as human ALOX15 (15-LOX1), human ALOX15B (15-LOX2), and human ALOX12 (platelet-type 12-LOX), exhibit allosteric properties. For these enzymes, the PUFA oxygenation products 13(*S*)-H(p)ODE [13(*S*)-hydro(pero)xyoctadecadienoic acid], 12(*S*)-H(p)ETE [12(*S*)-hydro(pero)xy-5Z,8Z,10E,14Z-eicosatetraenoic acid], and 14(*S*)-HDHA [14(*S*)-hydroxy-7Z,10Z,12E,16Z,19Z-docosapentaenoic acid] have been identified as allosteric effectors [135,136,137], and some synthetic fatty acid derivatives like OS (Figure 9) exhibit similar activities [138].

These compounds modified the kinetic properties of recombinant human ALOX15 in a way that was inconsistent with simple competitive inhibition of the enzyme. For instance, binding of 13(*S*)-HpODE changed the [*k_cat_*/K_M_]^AA^/[*k_cat_*/K_M_]^LA^ ratio over fivefold for human ALOX15 and threefold for human ALOX15B. 12(*S*)-HpETE modified this ratio only for human ALOX15B [135]. Human ALOX15B oxidizes AA with an efficiency [*k_cat_*/K_M_]^AA^ that overcomes the efficiency of LA oxygenation [*k_cat_*/K_M_]^LA^ by more than four times. In contrast, small levels of 13(*S*)-HODE (5 nM) lowered the ratio [*k_ca_*_t_/K_M_]^AA^/[*k_ca_*_t_/K_M_]^LA^ to about 2, and these data suggest a shift in substrate preference [139]. Isotopic effect experiments determined that the allosteric effectors lowered the hydrogen bond rearrangement step for the reaction with AA, increasing the relative importance of the hydrogen atom abstraction to the overall kinetics [136]. This was a significant result suggesting that allosteric binding may regulate substrate specificity by differentially affecting the rate constants of ALOX depending on the substrate.

In addition, 12(*S*)-HETE and 14(*S*)-HDPA [14(*S*)-hydroxy-7Z,10Z,12E,16Z,19Z-docosapentaenoic acid] were shown to decrease the production of 13(*S*),14(*S*)-epoxy-DHA from DHA by human ALOX15, which might be related to allosteric regulation of the enzyme. ALOX12 was also shown to be allosterically regulated by 14(*S*)-HDPA, but 12(*S*)-HETE had no effect [137]. A bicyclic pyrazoline derivative, initially named PKUMDL_MH_1001 [140,141], has been identified as a V-type activator of human ALOX15 [142]. This compound did not affect the K_M_ value but did increase the *k_cat_* AA and LA oxygenation. Since these compounds competed neither with active site inhibitors nor with allosteric effectors [142], a remote binding site that is different from the catalytic center was proposed. 3-O-acetyl-11-keto-b-boswellic acid, a naturally occurring pentacyclic triterpene (AKBA), did activate the cellular ALOX15 pathway upon binding to the inter-domain contact plane [143]. Unfortunately, we could not reproduce this activating effect using the purified native rabbit ALOX15 at 100 µM AA concentrations in the absence of any detergents. Thus, an increase in the cellular ALOX15 activity in the presence of 3-O-acetyl-11-keto-b-boswellic acid may also have been due to reasons other than allosteric regulation [144].

Summarizing the data obtained so far, one may conclude that the molecular basis of the allosteric character of ALOX isoforms, which is suggested by kinetic measurements, remains a matter of discussion. In fact, there are two major mechanistic scenarios: (i) the presence of an allosteric binding site at the enzyme outside the substrate binding pocket, which has not been characterized in detail, and (ii) protein dimerization, which will be discussed in the next section.

### 7.2. Inter- and Intra-Molecular Communication Mechanisms Responsible for Allosteric Properties of ALOX Isoforms

There were a number of attempts to characterize the binding sites for allosteric effectors in different ALOX isoforms, but none of these attempts provided convincing data. It has been shown that the interaction of ALOX15 with phosphatidylinositol bisphosphates strongly activates the fatty acid oxygenase activity in a Ca^2+^-dependent manner [145], but the molecular basis for this effect has not been clarified. Several partly solvent-exposed (more than 30%) nonpolar amino acids of the PLAT domain that are clustered at the inter-domain interface may be involved in the interaction with the hydrophobic core of membrane lipids or other molecule assemblies carrying a hydrophobic core. Di Venere at al. provided the first evidence that amphitropic enzymes like lipoxygenases experience opposite effects on their conformational heterogeneity when binding an active site ligand or interacting with biomembranes [146]. The presence of ETYA in the active site impairs motional flexibility of rabbit ALOX15, whereas the membrane-binding process induced an opposite effect [146]. Similar conclusions were drawn from data obtained by hydrogen-deuterium exchange (HDX) experiments for human ALOX15B. The presence of active site ligands (arachidonic acid or the isoform-selective inhibitor MLS000327069) resulted in structural rigidification of the α2 helix [13,86]. Moreover, comparison of the H/D exchange kinetics observed for cytosolic ALOX15B and a nanodisc-associated enzyme indicated that membrane association has a minimal effect on the structural dynamics of ALOX15B, suggesting differences in the allosteric behavior of human ALOX15 and ALOX15B. On the other hand, one may suggest that the binding of hydrophobic lipids or any other large hydrophobic molecule in the crevice between catalytic and PLAT domains may lead to the rearrangement of the original intramolecular contacts (Figure 4) that stabilize the α2 helix.

For pure rabbit ALOX15 SAXS measurements, the presence of 13(*S*)-HODE shifted the dynamic monomer–dimer equilibrium towards dimer formation [79]. Moreover, mutations at the inter-monomer interface (Trp181Glu+His585Glu, Leu183Glu+Leu192Glu) disturbed the hydrophobic interactions, which stabilize the ALOX15 dimer and induce a reduction in the catalytic activity [79]. These data suggest but do not really prove that ALOX15 may function as a transient enzyme dimer, in which one enzyme monomer (an allosteric monomer) may bind the effector molecule and then allosterically regulate the activity of the catalytic monomer. Using this dimeric enzyme model, it is possible to explain the pronounced differences in the IC_50_ values observed for the 2-arylindole derivative **1**-induced (Table 1, Figure 10A) inhibition of rabbit ALOX15 catalyzed oxygenation of AA and LA [121]. To support this hypothesis experimentally, we have shown that with LA as the substrate, compound **1** (Table 1) induced a marked decrease in *k_cat_*, but the K_M_-values remained unaffected [121]. On the other hand, for AA oxygenation, a marked decrease in both *k_cat_* and K_M_ values was observed. These data suggest a non-competitive inhibition of LA oxygenation with a K_i_ of 21.9 ± 0.5 nM. In contrast, for AA oxygenation, an uncompetitive mode of inhibition with a K_i_ of 0.95 ± 0.06 µM was determined. Taken together, these data suggest that binding of LA and binding of the inhibitor are independent processes and may occur at different sites. The inhibitor-induced reduction in the K_M_-value for AA oxygenation suggests an improved formation of the enzyme–substrate complex (ES^AA^) when the inhibitor is bound at the active site of the allosteric monomer. To support the plausibility of the dimer hypothesis as major reason for the allosteric properties of rabbit ALOX15, MD simulations of the enzyme substrate inhibitor dimeric complex were carried out. Here, we found that depending on the structure of the effector molecule bound at the active site of the allosteric monomer, the substrate may adopt different conformations in the substrate-binding pocket of the catalytic monomer, which either improves or impairs initial hydrogen abstraction [121,147].

To further support the hypothesis that dimer formation may be the main reason for the allosteric properties of human ALOX15, we compared the inhibitor efficiency of compound **1** (Table 1) on the LA and AA oxygenase activity of wildtype rabbit ALOX15 and its Leu183Glu+Leu192Glu double mutant, which exhibits a compromised dimerization [79]. In the case of the mutant enzyme, we found that the dose–response curves of LA and AA inhibition were very similar, with IC_50_ values of 4.95 ± 0.38 µM and 7.02 ± 0.91 µM, respectively (Figure 10B).

In contrast, for the wildtype enzyme, the dose–response curves and the IC_50_-values were very different (Figure 10A). These data suggest an allosteric character of the wildtype enzyme, which is strongly reduced when enzyme dimerization is prohibited (Figure 10C,D). Interestingly, the effect of the substrate selectivity of 2-arylindole derivative **1** (Table 1) was less pronounced when pig ALOX15 and the catalytic domain of rabbit ALOX15 were used (Figure 10E–G). Most mammalian ALOX15 pig ALOX15 (leukocyte-type 12-LOX) is an AA 12-lipoxygenating enzyme [18]. In this protein, Ile418 and Met419, which limit the depth of the substrate-binding cavity in rabbit ALOX15, are present as less space-filling Val residues. Hence, the volume of the substrate-binding pocket is bigger, and the substrate molecule has a higher degree of conformational flexibility when bound at the active site. In contrast to ALOX15, for which there is a monomer–dimer equilibrium in aqueous solutions, human ALOX12 (platelet-type 12-LOX) forms stable dimers. As for rabbit ALOX15, we identified two leucine residues (Leu183 and Leu187) as critical amino acids for protein dimerization [85]. The steady-state kinetics for the monomeric Leu183Glu+Leu187Glu double mutant were strongly impaired when compared with wildtype human ALOX12, which presents as an enzyme dimer. More interestingly, the inhibitory effect of the ML355 inhibitor against the Leu183Glu+Leu187Glu mutant monomer was almost completely abolished [85]. This discovery suggests that the conformational changes related to dimerization state could be also translated to active site residues and affect the inhibitor binding.

## 8. Conclusions

Mammalian ALOX15, ALOX15B, and ALOX12 orthologs are allosteric enzymes, but the molecular basis for this enzyme property is still a matter of discussion. In principle, there are several mechanistic scenarios to explain this kinetic property, but none of them have been convincingly confirmed. We recently suggested that enzyme dimerization may be the main reason for the allosteric properties of rabbit ALOX15 and that binding of an allosteric inhibitor in the substrate-binding pocket of the allosteric monomer may induce conformational alterations in the catalytic monomer, which result in enzyme inhibition. In other words, inter-monomer communication within the ALOX15 dimer may be considered the main reason for the allosteric character of the enzyme. Since enzyme dimerization has also been reported for human ALOX15B and human ALOX12, inter-monomer communication might also be considered the molecular basis for the allosteric properties of these two enzymes. Moreover, owing to the functional differences between mouse and human ALOX15 and ALOX15B, novel ALOX inhibitors should always be tested for their isoform specificity and for a lack of ortholog selectivity.

## Figures and Tables

**Figure 1 ijms-25-12058-f001:**
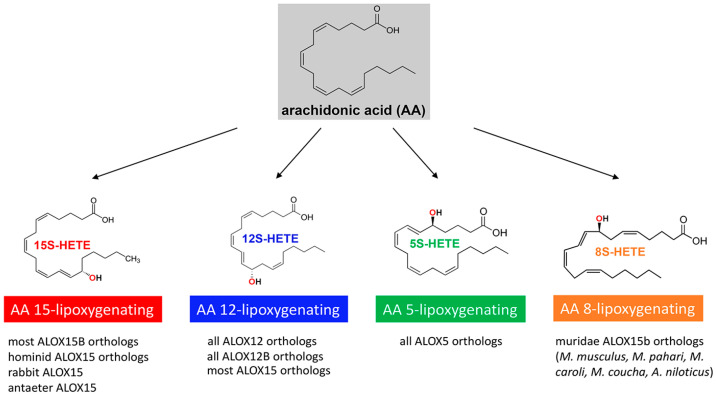
Classification of mammalian ALOX isoforms according to the conventional arachidonic acid-based nomenclature. According to the reaction specificity of arachidonic acid, oxygenation mammalian ALOX isoforms can be classified in four different groups: (i) AA 15-lipoxygenating enzymes, (ii) AA 12-lipoxygenating enzymes, (iii) AA 5-lipoxygenating enzymes, and (iv) AA 8-lipoxygenating enzymes. Typical representatives of these four classes are given. This nomenclature is outdated and should not be used any more since it ignores the evolutionary relationships between the enzymes. The use of the gene-based enzyme nomenclature is preferred.

**Figure 2 ijms-25-12058-f002:**
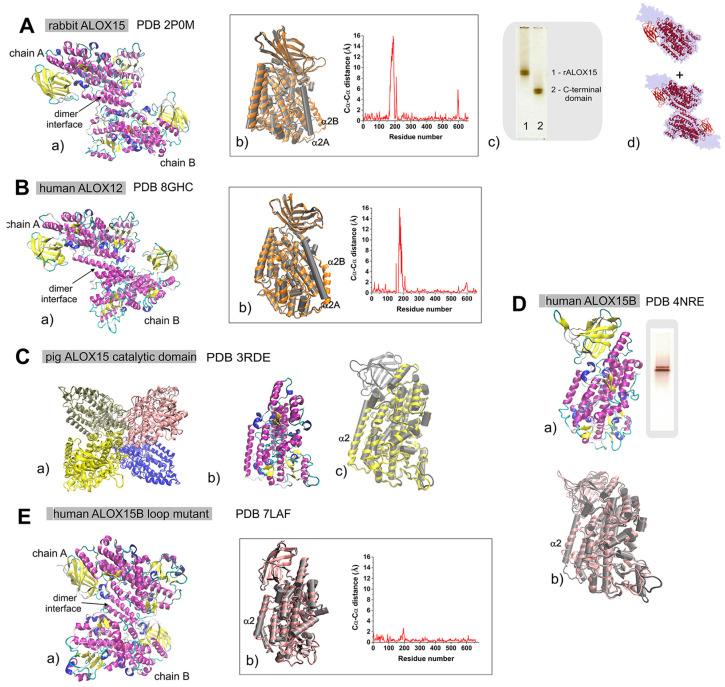
Structures of different mammalian ALOX. (**A**) Structure of rabbit ALOX15 (15-LOX1) (PDB 2P0M). (**a**) Ribbon diagram of the dimer structure: α-helices (purple), 3_10_-helices (blue), and β-sheets (yellow). (**b**) Overlay of crystal structures of monomer A (chain A, grey) and ligand-bound monomer B (chain B, orange). The Cα···Cα distances of monomers A and B versus residue number are plotted. (**c**) Native PAGE analysis of pure rabbit ALOX15 and its catalytic domain. (**d**) Overlay of the low-resolution structures [79] of pure native rabbit ALOX15 obtained in aqueous solutions (light blue) with the crystal structures of ALOX15 monomers and dimers. (**B**) Structure of human ALOX12 (platelet-type 12-LOX) (PDB 8HGC). (**a**) Ribbon diagram of the dimer structure: α-helices (purple), 3_10_-helices (blue), and β-sheets (yellow). (**b**) Overlay of the crystal structures of monomer A (chain A, grey) and monomer B (chain B, orange). The Cα···Cα distances of monomers A and B versus residue number are plotted. (**C**) Structure of pig ALOX15 (leukocyte-type 12-LOX) catalytic domain (PDB 3RDE). (**a**) Ribbon diagram of the tetramer structure in the enzyme crystals. For clarity, different chains are labelled by different colors. (**b**) Folding of the catalytic domain of pig ALOX15: α-helices (purple), 3_10_-helices (blue), and β-sheets (yellow). (**c**) Overlay of the crystal structures of rabbit ALOX15 (PDB 2P0M, monomer A, grey) and pig ALOX15 catalytic domain (PDB 3RDE, yellow). (**D**) Structure of human ALOX15B (15-LOX2) (PDB 4NRE) [12]. (**a**) Folding of human ALOX15B: α-helices (purple), 3_10_-helices (blue), and β-sheets (yellow) and native PAGE of the pure recombinant enzyme. (**b**) Overlay of crystal structures of rabbit ALOX15 (PDB 2P0M, monomer A, grey) and human ALOX15B (PDB 4NRE, pink). (**E**) Structure of the human ALOX15B (15-LOX2) loop mutant dimer complex with ligand (PDB 7LAF) [13]. (**a**) Ribbon diagram of the dimer structure: α-helices (purple), 3_10_-helices (blue), and β-sheets (yellow). (**b**) Overlay of the crystal structures of chain A (grey) and chain B (pink). The Cα···Cα distances versus residue number are plotted.

**Figure 3 ijms-25-12058-f003:**
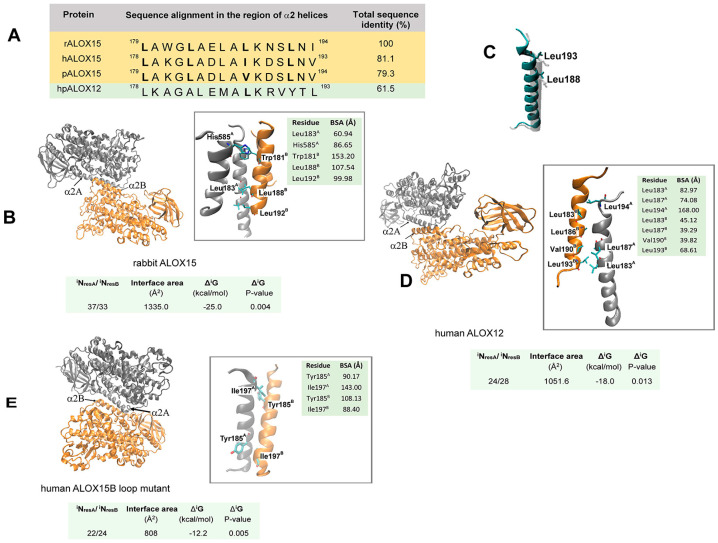
Sequence alignment and inter-monomer contacts in different mammalian ALOX isoforms. (**A**) Sequence alignment of selected mammalian ALOX. (**B**) Inter-monomer interface in the rabbit ALOX15 (15-LOX1) dimer (PDB 2P0M). The residues with the highest contribution are shown. (**C**) Structural overlay of α2 helices of rabbit (15-LOX1, gray transparent) and pig (leukocyte-type 12-LOX, cyan) ALOX15 orthologs. (**D**) Inter-monomer interface in human ALOX12 dimer (platelet-type 12-LOX, PDB 8GHC). The residues with the highest contribution are shown. (**E**) Inter-monomer interface in human loop mutantALOX15B (15-LOX2) dimer complex with ligand (PDB 7LAF).

**Figure 4 ijms-25-12058-f004:**
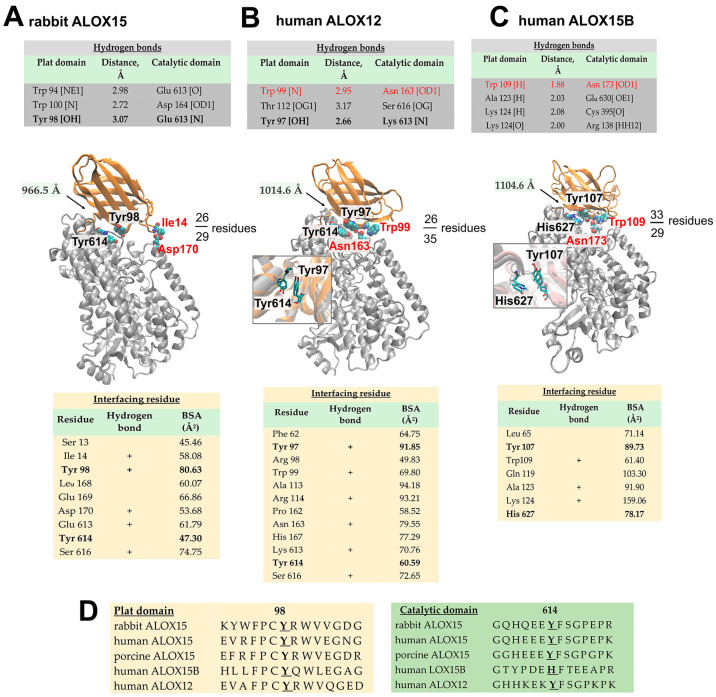
Interaction between the PLAT domain and the catalytic domain in mammalian ALOX isoforms. Interdomain interaction of the rabbit ALOX15 monomer A (PDB 2P0M) (**A**), the human ALOX12 (platelet-type 12-LOX) monomer A (PDB 8GHC) (**B**), and the human ALOX15B (15-LOX2, PDB 4NRE) (**C**). The numbers of amino acid residues of the two domains that contribute to the interdomain interface are indicated (e.g., 26/29 residues). Insets: Spatial orientation of Y98 and Y614 (rabbit ALOX15) relative to each other and structural overlays with the corresponding residues of human ALOX12 (**B**) and ALOX15B (**C**) are shown. The hydrogen-bonding network between the plat domain and the catalytic subunit according to results from PISA (www.ebi.ac.uk/msd-srv/prot_int, accessed on 20 March 2024) and the bonding distances of the hydrogen bridges are indicated. The interfacing residues that contributed the most are shown. (**D**) Multiple amino acid alignments of selected mammalian ALOXs surrounding Tyr98 and Tyr614R.

**Figure 5 ijms-25-12058-f005:**
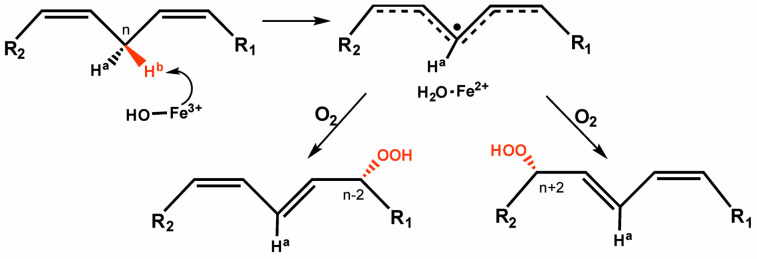
General mechanism of fatty acid oxidation by ALOX isoforms. The reaction is initiated by a hydrogen atom abstraction at carbon *n* followed by molecular oxygen insertion, which occurs antarafacially either at the *n* + 2 or the *n* − 2 carbons.

**Figure 6 ijms-25-12058-f006:**
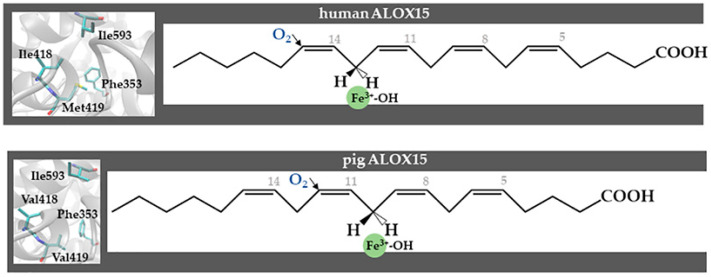
The triad concept, explaining the different reaction specificities of mammalian ALOX15 positional specificity. The triad determinants (Phe353, Ile418+Met419, Ile593 in human ALOX15) form the bottom of the substrate-binding pocket, and the space-filling residues at position 418 + 419 (Ile418+Met419) in human ALOX15 prevent a deeper penetration of the substrate fatty acids into the substrate-binding pocket. For this enzyme, arachidonic acid is bound at the active site in such a way that the bisallylic methylene C13 is localized in close proximity to the catalytic non-heme iron, and thus, molecular dioxygen is introduced at C15 of the substrate (**upper panel**). In pig ALOX15, less space-filling residue is localized at positions 418 and 419 (Val418+Val419), and thus, fatty acid substrates may penetrate deeper into the substrate-binding pocket. For this enzyme, the bisallylic methylene C10 is localized in close proximity to the non-heme iron, and thus, arachidonic acid 12-lipoxygenation becomes plausible (**lower panel**).

**Figure 7 ijms-25-12058-f007:**
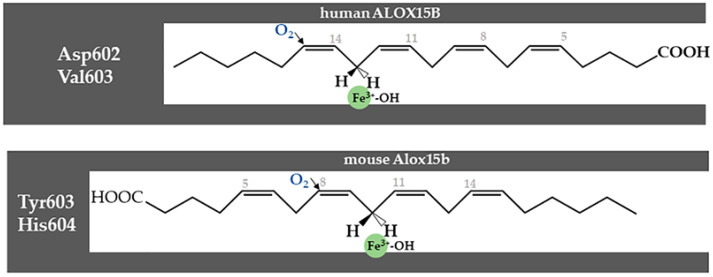
Structural concept explaining the different reaction specificities of human and mouse ALOX15B orthologs. In human ALOX15B, Asp602+Val603 (Jisaka positions) are localized at the bottom of the substrate-binding pocket, and the substrate fatty acids may penetrate the substrate-binding pocket with its methyl end ahead. Under these conditions, the bisallylic methylene C13 is localized in close proximity to the non-heme iron, and oxygen is introduced at C15 of arachidonic acid (15S-lipoxygenation). In mouse Alox15B, a Tyr+His motif is localized at the Jisaka positions. The positive charge of His604 may force inverse head-to-tail substrate alignment at the active site. Now, the bisallylic C10 of arachidonic acid is localized close to the non-heme iron, and oxygen is inserted at C8 (8S-lipoxygenation).

**Figure 8 ijms-25-12058-f008:**
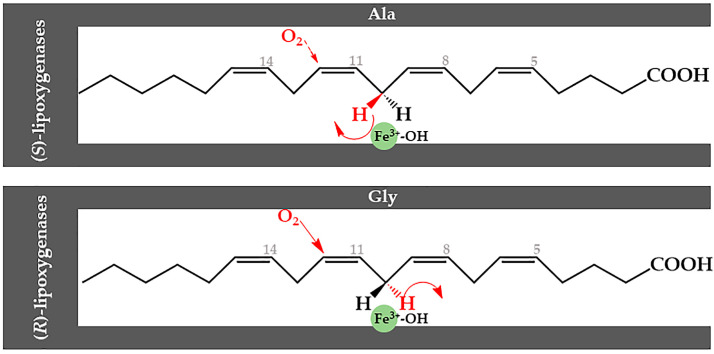
Structural concept explaining the enantioselectivity of *S*-lipoxygenating and *R*-lipoxygenating ALOX isoforms. According to the antarafacial character of the ALOX reaction, removal of the hydrogen atom that is localized above the plane determined by the double bonds of the fatty acid substrate is followed by oxygen insertion from below this plane. In this case, an oxygenation product with (*S*)-configuration is formed (S-lipoxygenases). On the other hand, when the hydrogen localized below the plane of the double bonds is abstracted, oxygenation products with (*R*)-configuration are synthesized (*R*-lipoxygenase).

**Figure 9 ijms-25-12058-f009:**
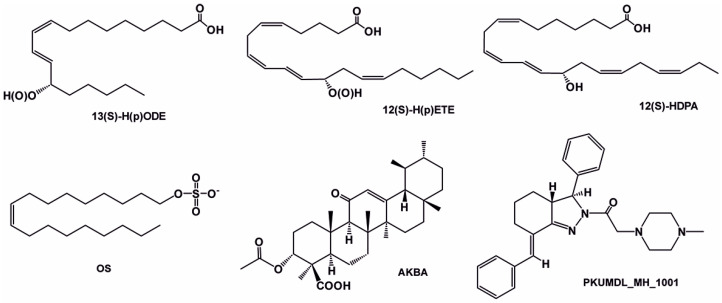
ALOX allosteric effectors.

**Figure 10 ijms-25-12058-f010:**
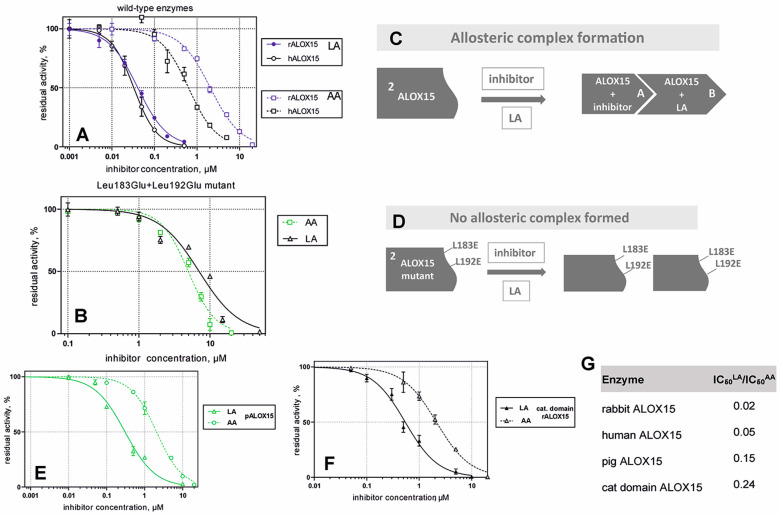
Dose−response curves for the inhibition of LA and AA oxygenase activities of mammalian ALOX15. The LA and AA oxygenase activities of pure recombinant rabbit and human ALOX15 (15-LOX1) (**A**), rabbit ALOX15 Leu183Glu+Leu192Glu double mutant (**B**), pig ALOX15 (leukocyte-type 12-LOX) (**E**), and catalytic domain of rabbit ALOX15 (**F**) were assayed in the presence or absence of compound **1** (Table 1), and the IC_50_^LA^/IC_50_^AA^ ratios are given (**G**). Substrate selectivity of ALOX allosteric effectors. (**C**,**D**) Schematic representation of the mechanism proposed for the inhibitor action.

**Table 1 ijms-25-12058-t001:** Selected highly potent ALOX15 inhibitors.

Comp.	Inhibitor Structure	IC_50_^AA^ nM	IC_50_^LA^ nM	Ref.
**1**	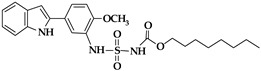	2060 ^(r)^ 660 ^(h)^	40 ^(r)^ 33 ^(h)^	[121]
**2**	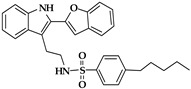	21 ^(r)^	6 ^(r)^	[123]
**3**	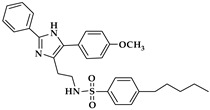	50 ^(r)^	14 ^(r)^	[124]
**4**	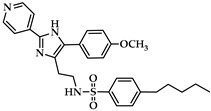	3211 ^(r)^	87 ^(r)^	[124]
**5**	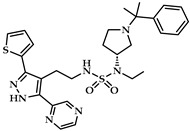	4 ^(r)^	3 ^(r)^	[125]
**6**	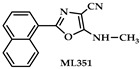	200 ^(h)^	n.d.	[119]

^r^ Rabbit recombinant ALOX15, ^h^ human recombinant ALOX15.

**Table 2 ijms-25-12058-t002:** Selected highly potent inhibitors of human ALOX15B and ALOX12.

Comp.	Inhibitor Structure	IC_50_^AA^ nM	Ref.
**7**	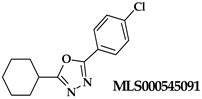	2600 ^a^	[127]
**8**	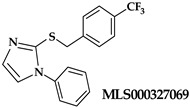	340 ^a^	[13]
**9**	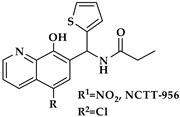	800 (R^1^) ^b^ 380 (R^2^) ^b^	[132]
**10**	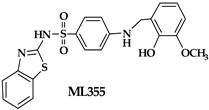	340 ^b^	[133]
**11**	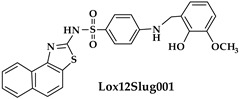	50 ^b^	[134]

^a^ Human ALOX15B (15-LOX2), ^b^ human ALOX12 (platelet-type 12-LOX).

## Data Availability

All data generated or analyzed during this study can be obtained upon request from I.I. or H.K.

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
