# Peer review of "Structural and Functional Biology of Mammalian ALOX Isoforms with Particular Emphasis on Enzyme Dimerization and Their Allosteric Properties"

_ijms, 2024, doi:10.3390/ijms252212058_

Round 1

Reviewer 1 Report

Comments and Suggestions for Authors

The manuscript provides a comprehensive review of the structural and functional biology of mammalian ALOX isoforms, with a particular focus on ALOX15, ALOX15B, and ALOX12. It delves into the allosteric properties of these enzymes and explores dimerization as a regulatory mechanism. While the review offers valuable insights, certain areas need more precise data and clear structural descriptions to strengthen the conclusions.

Major Comments:

1. Line 27: Exact Number of β-sheets and α-helices

The statement regarding the presence of "several parallel and anti-parallel β-sheets" is too vague. The exact number of β-sheets and α-helices should be provided for clarity and accuracy, as this level of detail is important in structural biology.

2. Line 281: Pocket Comparison

The claim that "the pocket is shallower than that of conformer B" requires quantification. The volume of the pockets should be reported, or a figure showing the superimposition of these two pockets should be provided to support the conclusion visually.

3. Line 343: Identity Percentage Clarification

Sixty percent identity is considered high, not moderate. The manuscript should reflect this understanding, as 60% identity typically suggests significant structural and functional similarities between proteins.

4. Line 366: Resolution Terminology

The term " The crystals diffract at a molecular resolution of at 1.89 Å." 1.89 Å resolution is atomic level. The correction should reflect that atomic, rather than molecular, resolution is achieved at this level.

5. Line 393: Sequence Identity and Structural Similarity

Thirty percent sequence identity often suggests similar overall protein structure. Therefore, there should be no surprise regarding the structural similarity between human ALOX15B and rabbit ALOX15. This point should be clarified to avoid confusion for the readers.

6. Figure 2: Color Labeling in Structures

The meaning of different colors in the labeled structures of Figure 2 should be explicitly stated. This will help readers understand what the color representations signify.

7. Figure 3: Hydrogen Bonding Distances

Hydrogen bonds typically form within the range of 2.7 Å to 3.3 Å. Residue pairs outside this range should not be classified as hydrogen bonds. The red-labeled residue pairs in the table should be defined, and the meaning of the 26/29 AA should be clarified.

8. Section 5: Structural Figures Needed

This section would greatly benefit from a figure illustrating the structural basis for the reaction specificity of these enzymes. Relying solely on text makes it difficult for readers to visualize and understand the important structural details.

Minor Comments:

1. Figure 1: Structural Representation Consistency

The space-filling models in Figure 1 should be replaced with ribbon diagrams to help readers better visualize how the multimer interfaces form, especially in relation to secondary structures. Additionally, ensure that all structural representations across figures (Figure 1A(b), B(b), C(b), C(c), D(b), D(c)) are consistent—either use ribbon diagrams or cylinders uniformly.

2. Line 337: r.m.s.d. for Structural Similarity

The manuscript should provide the root-mean-square deviation (r.m.s.d.) values to quantify the structural similarity between ALOX12 structures. This will offer readers a more objective measure of structural conservation.

3. Line 383: Clarification via Commas

A comma is needed after "ALOX15" in the sentence "As for the catalytic domain of pig ALOX15 the α2 helixes may not play a major role in protein oligomerization." This will make the sentence clearer to the reader. Similarly, in Line 409, "In the dimeric complex two structurally different monomers A and B interact with each other mainly via their α2 helices," a comma is required between "complex" and "two." The manuscript should be carefully proofread for similar punctuation corrections.

4. Line 1088: Consistent Terminology

The manuscript alternates between "porcine ALOX15" and "pig ALOX15." It is essential to use consistent terminology throughout the text. Choose either "porcine" or "pig" and apply it consistently to avoid confusion.

Author Response

Dear editor

We should like to thank the two reviewers for critical reading of our ms and for valuable advice, which have been considered for structuring the revised version of the ms. To address the concerns of the reviewers we rephrased large parts of the ms and corrected typos. Moreover, we provided additional information requested by the reviewer, restructured the list of refences (reduced the number of self-citations) and improved the quality of figures.

To speed up the review process of the revised version of the ms we resubmit an unmarked version of the revised ms (unmarked-rev-ms.docx), which can be used for publications, and a marked version (markedrev-ms.docx), in which the alterations introduced during revision are clearly labeled in yellow.

The critical comments of the reviewers have been addressed during revision as follows on the point-to-point basis.

Reviewer 1

Major comets

Comment 1 of reviewer 1: The manuscript provides a comprehensive review of the structural and functional biology of mammalian ALOX isoforms, with a particular focus on ALOX15, ALOX15B, and ALOX12. It delves into the allosteric properties of these enzymes and explores dimerization as a regulatory mechanism. While the review offers valuable insights, certain areas need more precise data and clear structural descriptions to strengthen the conclusions.

Response of authors: We thank reviewer 1 for overall positive evaluation of the ms and for the constructive comments, which have been considered for preparation of the revised version of the ms.

Comment 2 of reviewer 1: Line 275: Exact Number of β-sheets and α-helices: The statement regarding the presence of "several parallel and anti-parallel β-sheets" is too vague. The exact number of β-sheets and α-helices should be provided for clarity and accuracy, as this level of detail is important in structural biology.

Response of authors: The exact number of β-sheets and α-helices is provided in the revised version of the ms (line 275). This sentence now reads: The N-terminal PLAT (polycystin-1-lipoxygenase α-toxin) domain consists of 8 parallel and anti-parallel beta-sheets. The C-terminal catalytic domain involves the catalytically active non-heme iron and consists of 20 alpha-helices, 10 small 310-helices, 9 beta-sheets in one protein molecule. The other monomer in the ALOX15 dimer involves 19 alpha-helices, 10 small 310-helices and 6 beta-sheets.

Comment 3 of reviewer 1: 2. Line 281: Pocket Comparison. The claim that "the pocket is shallower than that of conformer B" requires quantification. The volume of the pockets should be reported, or a figure showing the superimposition of these two pockets should be provided to support the conclusion visually.

Response of authors: The description of the overall structure of the substrate binding pocket of the two enzyme monomers originates from ref 8 and this is clearly indicated in the revised version of the ms (line 284). We did not quantitatively evaluate the size of the substrate binding pockets of the two monomers on the basis of the X-ray diffraction data and thus, we cannot provide more quantitative data.

Comment 4 of reviewer 1: Line 343: Identity Percentage Clarification. Sixty percent identity is considered high, not moderate. The manuscript should reflect this understanding, as 60% identity typically suggests significant structural and functional similarities between proteins.

Response of authors: We do agree with the reviewer that in general 60% amino acid identity suggests significant structural and functional similarities between proteins. However, if one compares the degrees of amino acid conservation among different ALOX-isoforms a 60% amino acid identity is rather low. In fact, mammalian ALOX15 orthologs share 80-90% amino acid identity and similar values were obtained for mammalian ALOX15B and ALOX12 orthologs. On the other hand, human ALOX15 only shares 60% amino acid identity with human ALOX12. We modified the text accordingly (line 355).

Comment 5 of reviewer 1: Line 366: Resolution Terminology. The term " The crystals diffract at a molecular resolution of at 1.89 Å." 1.89 Å resolution is atomic level. The correction should reflect that atomic, rather than molecular, resolution is achieved at this level.

Response of authors: We follow the advice of the reviewer and corrected our terminology. In the revised version of the ms the corresponding sentence (line 385) reads: The protein crystals diffract at atomic resolution of 1.89 Å …

Comment 6 of reviewer 1: Line 393: Sequence Identity and Structural Similarity. Thirty percent sequence identity often suggests similar overall protein structure. Therefore, there should be no surprise regarding the structural similarity between human ALOX15B and rabbit ALOX15. This point should be clarified to avoid confusion for the readers.

Response of authors: The reviewer is correct when stating that 30% amino acid identity often suggests similar overall structures and functions. However, if one compares mammalian ALOX isoforms 40% amino acid identity is rather low. Nevertheless, all mammalian ALOX isoforms X-rayed so far have similar 3D structures. In response to the critical remark of the reviewer we rephrased the corresponding sentence (line 406), which now reads: Human ALOX15B shares 40% amino acid sequence identity with rabbit ALOX15 and the overall 3D structures of the two enzymes are quite similar.

Comment 7 of reviewer 1: Figure 2: Color Labeling in Structures The meaning of different colors in the labeled structures of Figure 2 should be explicitly stated. This will help readers understand what the color representations signify.

Response of authors: We agree with the reviewer that extensive color labeling may impair legibility of complex images and this might be the case for the original version of Fig. 2.  For the revised version of this image, we minimized color labeling and focused on the structural interactions, which are relevant for ALOX dimerization.

Comment 8 of reviewer 1: Figure 3: Hydrogen Bonding Distances. Hydrogen bonds typically form within the range of 2.7 Å to 3.3 Å. Residue pairs outside this range should not be classified as hydrogen bonds. The red-labeled residue pairs in the table should be defined, and the meaning of the 26/29 AA should be clarified.

Response of authors: The reviewer is correct when stating that the bonding distance of hydrogen bonds must not exceed 3.3 Å. We restructured Fig. 3 accordingly and explained the meaning of the figure element “26/29 AA” in the figure legend (line 521).

Comment 9 of reviewer 1: Section 5: Structural Figures Needed. This section would greatly benefit from a figure illustrating the structural basis for the reaction specificity of these enzymes. Relying solely on text makes it difficult for readers to visualize and understand the important structural details.

Response of authors: We thank the reviewer for this constructive comment and prepared three additional images. These images visualize in a schematic way the mechanistic basis for the different reaction specificities of human and pig ALOX15 (Fig. 5, line 649-660), of human and mouse ALOX15B (Fig. 6, line 745) and S- vs. R-lipoxygenases (Fig. 7, line 851). For Fig. 5 we attempted to structure an image that was based on the X-ray coordinates of the two enzymes. These attempts were not successful since it is difficult to visualize spatial differences in the architectures of the substrate binding pockets in a two-dimensional image. Moreover, X-ray coordinates for enzyme-substrate complexes are currently not available for the two enzymes and structural models are not reliable. For the enzymes shown in Fig. 6 + 7 this is also the case.

Comment 10 of reviewer 1: 1. Figure 1: Structural Representation Consistency. The space-filling models in Figure 1 should be replaced with ribbon diagrams to help readers better visualize how the multimer interfaces form, especially in relation to secondary structures. Additionally, ensure that all structural representations across figures (Figure 1A(b), B(b), C(b), C(c), D(b), D(c)) are consistent—either use ribbon diagrams or cylinders uniformly.

Response of authors: In response to this comment of the reviewer we restructured Fig. 1 and the following alterations were introduced: i) We replaced the space-filling models in Fig. 1 with ribbon diagrams. ii) The secondary structural elements are clearly indicated in the revised version of Fig. 1. iii) We removed the hexamer structure of human ALOX15B (panel D-a) but introduced an additional panel   visualizing the dimerization of the human ALOX15B loop mutant.

Comment 11 of reviewer 1: 2. Line 337: r.m.s.d. for Structural Similarity. The manuscript should provide the root-mean-square deviation (r.m.s.d.) values to quantify the structural similarity between ALOX12 structures. This will offer readers a more objective measure of structural conservation.

Response of authors: This is a good idea and we modified Fig.1 accordingly. The images now involve diagrams showing the Cα - Cα distances of monomers A and B over the residue numbers to quantify the degree of structural similarity.

Comment 12 of reviewer 1: 3. Line 383: Clarification via Commas: A comma is needed after "ALOX15" in the sentence "As for the catalytic domain of pig ALOX15 the α2 helixes may not play a major role in protein oligomerization." This will make the sentence clearer to the reader. Similarly, in Line 409, "In the dimeric complex two structurally different monomers A and B interact with each other mainly via their α2 helices," a comma is required between "complex" and "two." The manuscript should be carefully proofread for similar punctuation corrections.

Response of authors: We corrected the text as suggested by the reviewer. We did our best to avoid spelling and grammar errors but since neither of the authors is a native English speaker, correct punctuation is rather difficult. The final ms was proof read by a native speaker but this is no guarantee.

Comment 13 of reviewer 1: 4. Line 1088: Consistent Terminology. The manuscript alternates between "porcine ALOX15" and "pig ALOX15." It is essential to use consistent terminology throughout the text. Choose either "porcine" or "pig" and apply it consistently to avoid confusion.

Response of authors: We followed the advice of the reviewer and used the term “pig” throughout the ms.

Reviewer 2 Report

Comments and Suggestions for Authors

1.       In line 42, ‘at a molecular resolution of’ should be changed to ‘ at the molecular resolution of’

2.       In line 52, ‘were recently obtained for’ should be changed to ‘were recently obtained in’

3.       In line 66, ‘share a 38%’ should delete ‘a’

4.       In line 94, ‘ALOX5is of particular importance’  add a space between ‘ALOX5’ and ‘is’, and delete ‘of’.

5.       In Introduction section, authors descript the lipoxygenase nomenclature into four different types, it’s better to provide a figure with chemical structure to descript the substrate and product for each type. It’s easy to understand what’s the difference between different types.

6.       In line 415, ‘Side-directed mutagenesis’ to ‘site-directed mutagenesis’

7.       In this paper, PDB ID labeled is not consistent, some PBD XXXX, some PDB ID XXXX, some PDB: XXXX

8.       It’s hard for me to understand Figure 3, if you want to show the interface between two proteins, you do not need  to show whole structures, just shown interaction region is fine.

9.       Please provide corresponding figure for each point, otherwise it’s hard to understand your description. For example, in section 5, Mechanism of ALOX, there is too much description, only one figure, it hard for audience to understand it.

Comments on the Quality of English Language

The quality of English need to improve to let reader understand more easier.

Author Response

Dear editor

We should like to thank the two reviewers for critical reading of our ms and for valuable advice, which have been considered for structuring the revised version of the ms. To address the concerns of the reviewers we rephrased large parts of the ms and corrected typos. Moreover, we provided additional information requested by the reviewer, restructured the list of refences (reduced the number of self-citations) and improved the quality of figures.

To speed up the review process of the revised version of the ms we resubmit an unmarked version of the revised ms (unmarked-rev-ms.docx), which can be used for publications, and a marked version (markedrev-ms.docx), in which the alterations introduced during revision are clearly labeled in yellow.

The critical comments of the reviewers have been addressed during revision as follows on the point-to-point basis.

Reviewer 2

Comment 1 of reviewer 2: 1. In line 42, ‘at a molecular resolution of’ should be changed to ‘ at the molecular resolution of’

Response of authors: We corrected this sentence, which now reads: … at the atomic resolution of 2.4 Å

Comment 2 of reviewer 2: In line 51, ‘were recently obtained for’ should be changed to ‘were recently obtained in’

Response of authors: In response to this comment, we have modified this sentence. Now it reads (line 51): “In addition, several high resolution cryo-EM structures of different oligomeric forms of human ALOX12 (platelet-type 12-LOX according to the old nomenclature; PDB 8GHB to 8GHE) have recently been published [14]”.

Comment 3 of reviewer 2: In line 66, ‘share a 38%’ should delete ‘a’

Response of authors: We corrected this sentence as suggested by the reviewer (line 66).

Comment 4 of reviewer 2: In line 94, ‘ALOX5is of particular importance’  add a space between ‘ALOX5’ and ‘is’, and delete ‘of’.

Response of authors: We apologize for this misprint and we have added a space between ‘ALOX5’ and ‘is’ (line 94).

Comment 5 of reviewer 2: In Introduction section, authors descript the lipoxygenase nomenclature into four different types, it’s better to provide a figure with chemical structure to descript the substrate and product for each type. It’s easy to understand what’s the difference between different types.

Response of authors: In response to this comment of the reviewer we modified the corresponding chapter of the Introduction section and structured an additional image (revFig1). 1to visualize the different product patterns of mammalian 12- and 15-lipoxygenating ALOX15 orthologs as well as human and mouse ALOX15B orthologs.

Comment 6 of reviewer 2: In line 415, ‘Side-directed mutagenesis’ to ‘site-directed mutagenesis’

Response of authors: We corrected this typo.

Comment 7 of reviewer 2: In this paper, PDB ID labeled is not consistent, some PBD XXXX, some PDB ID XXXX, some PDB: XXXX

Response of authors: We apologize for this typo and corrected it.

Comment 8 of reviewer 2: It’s hard for me to understand Figure 3, if you want to show the interface between two proteins, you do not need to show whole structures, just shown interaction region is fine.

Response of authors: With all due respect for the expertise of the reviewer we disagree with this statement. If we only show the interacting regions non-expert readers may have problems to localize these regions in the global structures. This may not be the case for ALOX specialists who a familiar with the global ALOX structures. However, since we expect that this paper is also read non-ALOX experts we would like to keep Fig. 3 as it is.

Comment 9 of reviewer 2: Please provide corresponding figure for each point, otherwise it’s hard to understand your description. For example, in section 5, Mechanism of ALOX, there is too much description, only one figure, it hard for audience to understand it.

Response of authors. We thank the reviewer for this constructive comment and prepared three additional images. These images visualize in a schematic way the mechanistic basis for the different reaction specificities of human and pig ALOX15 (Fig. 5), of human and mouse ALOX15B (Fig. 6) and S- vs. R-lipoxygenases (Fig. 7). For Fig. 5 we attempted to structure an image that was based on the X-ray coordinates of the two enzymes. These attempts were not successful since it is difficult to visualize spatial differences in the architectures of the substrate binding pockets in a two-dimensional image. Moreover, X-ray coordinates for enzyme-substrate complexes are currently not available for the two enzymes and structural models are not reliable. For the enzymes shown in Fig. 6 + 7 this is also the case.

We hope that we have adequately addressed the comments of the reviewers and that the revised version of the ms is now acceptable for publication in iJMS.

Sincerely,

  1. Kuhn, I Ivanov

Round 2

Reviewer 1 Report

Comments and Suggestions for Authors

well done.

Reviewer 2 Report

Comments and Suggestions for Authors

This version looks much better. Thanks for your reply.